# Sulfur stabilizing metal nanoclusters on carbon at high temperatures

Peng Yin[1,5], Xiao Luo[1,2,5], Yanfu Ma[1,5], Sheng-Qi Chu [3], Si Chen[1], Xusheng Zheng[4], Junling Lu [1✉], Xiao-Jun Wu [1,2✉] & Hai-Wei Liang [1✉]

Supported metal nanoclusters consisting of several dozen atoms are highly attractive for heterogeneous catalysis with unique catalytic properties. However, the metal nanocluster catalysts face the challenges of thermal sintering and consequent deactivation owing to the loss of metal surface areas particularly in the applications of high-temperature reactions. Here, we report that sulfur—a documented poison reagent for metal catalysts—when doped in a carbon matrix can stabilize ~1 nanometer metal nanoclusters (Pt, Ru, Rh, Os, and Ir) at high temperatures up to 700 °C. We find that the enhanced adhesion strength between metal nanoclusters and the sulfur-doped carbon support, which arises from the interfacial metal-sulfur bonding, greatly retards both metal atom diffusion and nanocluster migration. In catalyzing propane dehydrogenation at 550 °C, the sulfur-doped carbon supported Pt nanocluster catalyst with interfacial electronic effects exhibits higher selectivity to propene as well as more stable durability than sulfur-free carbon supported catalysts.

[1] Hefei National Laboratory for Physical Sciences at the Microscale, School of Chemistry and Materials Sciences, University of Science and Technology of China, Hefei, China. [2] Synergetic Innovation of Quantum Information & Quantum Technology, CAS Key Laboratory of Materials for Energy Conversion, and CAS Center for Excellence in Nanoscience, University of Science and Technology of China, Hefei, Anhui, China. [3] Beijing Synchrotron Radiation Facility, Institute of High Energy Physics, Chinese Academy of Sciences, Beijing, China. [4] National Synchrotron Radiation Laboratory, University of Science and Technology of China, Hefei, Anhui, P. R. China. [5] These authors contributed equally: Peng Yin, Xiao Luo, Yanfu Ma. ✉email: junling@ustc.edu.cn; xjwu@ustc.edu.cn; hwliang@ustc.edu.cn

Supported metal catalysts occupy a pivotal position in diverse technological applications, especially for the sustainable production of chemicals and fuels[1–3]. The increase of metal dispersion on supports by downsizing metals to nanoscale is of significance for maximizing the metal atom utilization and consequently increasing the activity normalized to mass[4–7]. The extreme metal utilization can be achieved when the particle size approaches 1 nm (that is, nanoclusters of several dozen atoms), as almost all metal atoms at this size are accessible for catalysis[8]. On the other hand, the electronic properties of metals are also highly dependent on their size particularly below 2 nm, due to the quantum size effect[9], which has been shown to be crucial for enabling the metal catalysts with high intrinsic activity and unexpected selectivity[6,10–12].

However, an inherent problem of metal nanoclusters for catalysis applications is their well-documented thermodynamic instability[13–15], as metal species tend strongly to grow into larger crystallites due to the sharply increased surface energy with the decrease of particle size, through the particle migration and coalescence (PMC) and/or Ostwald ripening (OR) processes[16,17]. Such metal sintering inevitably leads to the loss of active surface area and thus the catalyst deactivation, especially for high-temperature catalytic reactions. Considerable efforts have, therefore, been devoted in the past decade to developing different physical and chemical approaches for suppressing metal nanocluster sintering at high temperatures. The most adopted method is to elaborately tune the spatial arrangement of metals and supports at the nanoscale for constructing the physical barriers against sintering, such as the maximization of inter-particle spacing by filling metal nanoclusters in the channels of mesoporous silica and carbon supports[18,19] and the encapsulation of metal nanoclusters with porous nanoshells (e.g., zeolite and metal oxides)[20–23]. Albeit conceptually efficient, these methods may cause decreased active surface area and increased mass-transfer resistance, which can considerably lower the overall performance of the catalysts. Recently, some nanostructured metal oxide supports have been proven to be capable of stabilizing metal species at high temperature through metal–support bonding (for example, the Pt–O–Ce bonding in Pt/CeO$_2$ catalyst) in oxidative atmospheres[24–27]. In another breakthrough work, Hu et al. reported the high-temperature shockwave method to stabilize metal single atoms on carbon, C$_3$N$_4$, and TiO$_2$ supports by forming the metal-defect bonds[28].

Here, we demonstrate that sulfur—a traditional poison reagent for metal catalysts[29]—when doped in a carbon matrix can efficiently stabilize ~1 nm metal nanoclusters (Pt, Ru, Rh, Os, and Ir) against thermal sintering at high temperatures up to 700 °C in a reductive atmosphere. Spectroscopic characterization and density functional theory (DFT) calculations reveal that the enhanced adhesion strength at the metal/sulfur-doped carbon (S–C) interface, arising from the strong and thermally stable metal-sulfur bonding, greatly suppresses the metal nanocluster sintering kinetics by retarding both metal atom diffusion (OR path) and nanoparticle migration (PMC path). Moreover, the prepared Pt nanocluster catalysts exhibit high activity, selectivity, and long reaction lifetimes in the high-temperature reaction of propane dehydrogenation (PDH) to propylene at 550 °C, demonstrating the application potentials of the metal nanocluster catalysts for industrially relevant catalysis under realistic technical conditions.

## Results

### Sinter resistance of S–C supported nanocluster catalysts. Figure 1 shows our concept of suppressing metal migration/coalescence and ripening to achieve the sinter-resistant metal nanocluster catalysts based on the strong chemical interaction

between metal and doped sulfur atoms in the S–C supports. We first prepared the S–C support by cobalt-assisted carbonization of molecular precursors (that is, 2,2'-bithiophene) with SiO$_2$ nanoparticles as templates, according to our previously reported works[30,31]. The S–C support with structural features of a high sulfur content (~14 wt %) and a large specific surface area (>1200 m$^2$g$^{-1}$) then served as the support for loading of Pt nanoclusters (1 wt %) by the wet-impregnation of H$_2$PtCl$_6$ and subsequent H$_2$-reduction at 700 °C. High-angle annular dark-field scanning transmission electron microscopy (HAADF-STEM) measurements of the S–C supported Pt nanocluster (Pt/S–C) catalyst showed that numerous Pt nanoclusters with size in the range of 0.8–1.3 nm (mean size, 1.0 nm) homogeneously distributed over the whole carbon matrix, without any large particles or nanocluster aggregates (Fig. 2a and Supplementary Fig. 1). Aberration-corrected HAADF-STEM image with atomic resolution indicates that the Pt nanoclusters are highly crystalline and mainly enclosed by (111) and (200) crystal planes of face-centered cubic Pt (Supplementary Fig. 2). Note that there were still considerable sulfur atoms survived in the carbon carrier even after the high-temperature H$_2$-reduction at 700 °C during catalyst preparation (Fig. 3a). We further surmise that these thermally stable sulfur atoms are crucial for stabilizing metal nanoclusters at high temperatures.

Owing to the dramatically increased surface energy of Pt particle below 2 nm compared to their bulk counterparts, the small-sized Pt nanoparticles would be highly mobile and have strong tendency of migration/coalescence and ripening even on high-surface-area supports[17]. We then studied the high-temperature thermal stability of the Pt/S–C catalyst by treating it at 700 °C for a long period time of 600 min under flowing 5% H$_2$/Ar. Remarkably, no any aggregation or overgrowth of Pt nanoclusters were found by HAADF-STEM observations with a broad vision after the harsh thermal treatment (Fig. 2b and Supplementary Fig. 3). Moreover, the abundant sulfur-containing sites and highly porous structure of S–C enabled us to prepare the Pt nanocluster catalysts in a high loading of 5 wt%, which also exhibited outstanding sinter-resistant properties (Fig. 2c, d). Note that considerable Pt nanoparticle growth was observed in the N-doped[32] and thiolated carbon nanotube supports[33] upon the high-temperature treatments. Although some sulfur-doped carbon supports have been reported to stabilize Pt nanoparticles of 3–5 nm up to 600 °C[34–36], to our knowledge, so far few works have reported that metal clusters of ~1 nm could be stabilized on carbon supports up to 700 °C. We also found obvious Pt sintering on the mesoporous N-doped carbon supports at 700 °C (Supplementary Fig. 4), which was attributed to the significantly decreased nitrogen content from 3.01 to 1.44 at% after the sintering test, as revealed by the X-ray photoelectron spectroscopy (XPS) measurements (Supplementary Fig. 5). Differently, the sulfur content of Pt/S–C kept almost intact after the same sintering test (Fig. 3a and Supplementary Table 1). The highly structural stability of the S–C supports with the presence of Pt at high temperatures is the premise of preventing nanoclusters from sintering. It is noteworthy that the S–C supports would burn out at high temperatures under oxygen atmosphere. Therefore, the S–C supported metal nanocluster catalysts maybe only applicable to the high-temperature reactions under reductive atmospheres.

To experimentally verify the crucial role of the doped sulfur atoms in stabilizing metal nanoclusters, we removed sulfur from the S–C supports substantially by thermally treating it at an extremely high temperature of 1100 °C in 5% H$_2$/Ar for 2 h. Both the mesoporous structure and high surface area of S–C maintained invariable after the desulfurization treatment (Supplementary Fig. 6), while the sulfur content dramatically decreased to only 0.18 wt% as determined by the elemental

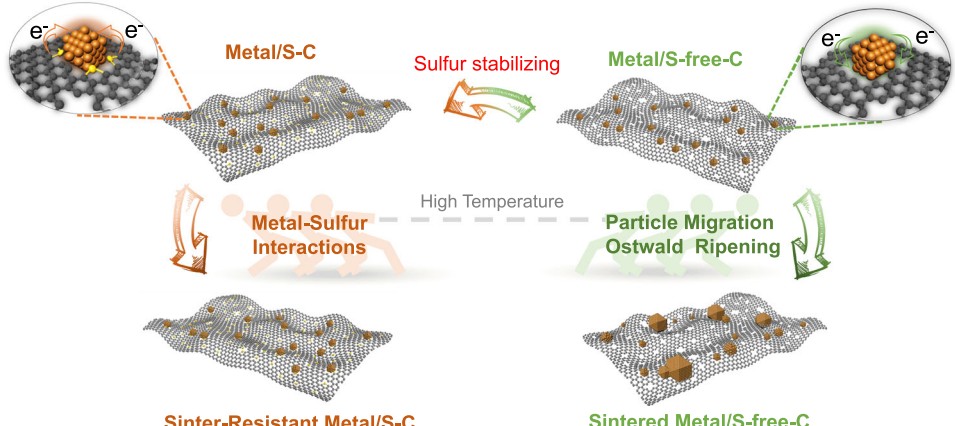

**Fig. 1 Schematic illustration of sulfur-stabilizing method.** The S–C supported metal nanocluster catalysts are resistant to sintering owing to the strong metal-sulfur interaction, whereas S-free-C supported catalysts easily aggregate into larger particles in high-temperature reactions.

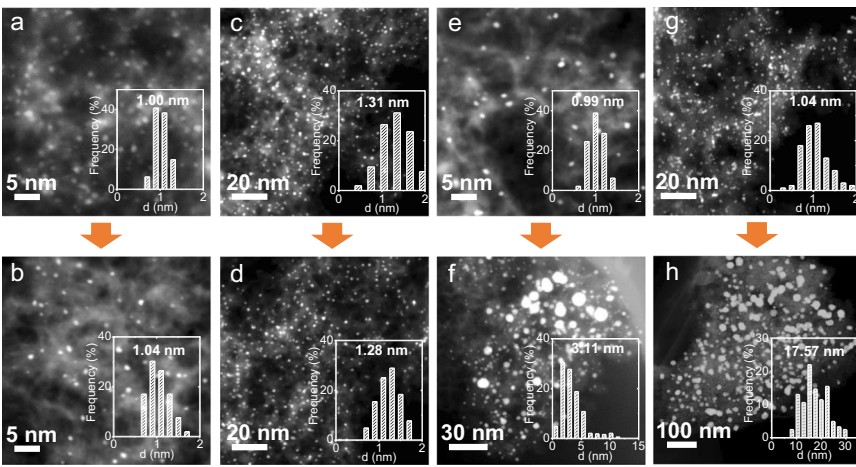

**Fig. 2 Sintering tests. a–h** HADDF-STEM images of the catalysts before (**a**, **c**, **e**, **g**) and after (**b**, **d**, **f**, **h**) sintering tests, including 1 wt% Pt/S–C (**a**, **b**), 5 wt% Pt/S–C (**c**, **d**), 1% Pt/S-free-C (**e**, **f**), and commercial Pt/C catalyst (**g**, **h**). Sintering test condition: 5% $H_2$/Ar, 700 °C, 600 min.

analysis (Supplementary Table 2). The desulfurized support is marked as S-free-C for the following discussion. Next, the same impregnation method was applied to load Pt on the desulfurized support (that is, S-free-C). For a low Pt loading of 1.0 wt%, we also obtained uniformly dispersed Pt nanoclusters that have similar size of ~1 nm with Pt/S–C on the S-free-C support (Fig. 2e). After the long-term thermal treatment at 700 °C, in sharp contrast, the S-free-C supported Pt nanoclusters experienced obvious sintering as indicated by HAADF-STEM observations that clearly showed a broad particle size distribution ranging from a few to around ten nanometers (Fig. 2f and Supplementary Fig. 7). Energy dispersive spectroscopic (EDS) elemental mapping further affirmed the metal sintering and ultralow sulfur content (<0.01 at %) for Pt/S-free-C (Supplementary Fig. 8). In the case of commercial Pt/C catalyst, as expected, we also observed the severe aggregation and overgrowth of Pt nanoparticles after the thermal treatment (Fig. 2g and h), due to the lack of strong metal-carbon support interactions.

To demonstrate the wide applicable scope of our synthetic strategy, we additionally prepared another four platinum-group metal (PGM) nanocluster catalysts and studied their anti-sintering properties under the same conditions, including Ru, Rh, Os, and Ir (1.0 wt% loading for all cases). HAADF-STEM images showed that these catalysts were composed of highly dispersed metal nanoclusters with mean sizes in the range of

0.46–0.82 nm (Supplementary Fig. 9). Low-magnification HAADF-STEM images with large views further indicated the homogeneity of these metal nanoclusters (Supplementary Fig. 10). Importantly, these metal nanoclusters also exhibited remarkable sinter-resistant properties without visible particle growth and agglomeration during the long-term thermal treatment at 700 °C for 10 h (Supplementary Figs. 9, 10) and the sulfur still existed stably in the carbon matrix (Supplementary Fig. 11 and Supplementary Table 1). The S-free-C supported Ru, Rh, Os, or Ir nanoclusters, in marked contrast to S–C supported ones, experienced different degrees of sintering; the average size of Ru, Rh, Os, and Ir nanoclusters on the S-free-C supports increased greatly from below 1 nm to 13.74, 7.96, 5.24, and 6.12 nm, respectively (Supplementary Figs. 12–14). Moreover, the sulfur-stabilization method was further extended to prepare stable bimetallic and multi-metallic cluster catalysts (Supplementary Fig. 15) and even to prepare sintering-resistant nanocluster catalysts with sulfur-doped $TiO_2$ support (Supplementary Figs. 16, 17).

**Interfacial electronic effects between Pt and S–C.** We then performed X-ray absorption near edge structure (XANES) and extended X-ray absorption fine structure (EXAFS) analyses to verify the strong metal/sulfur interactions. The Pt/S–C catalyst

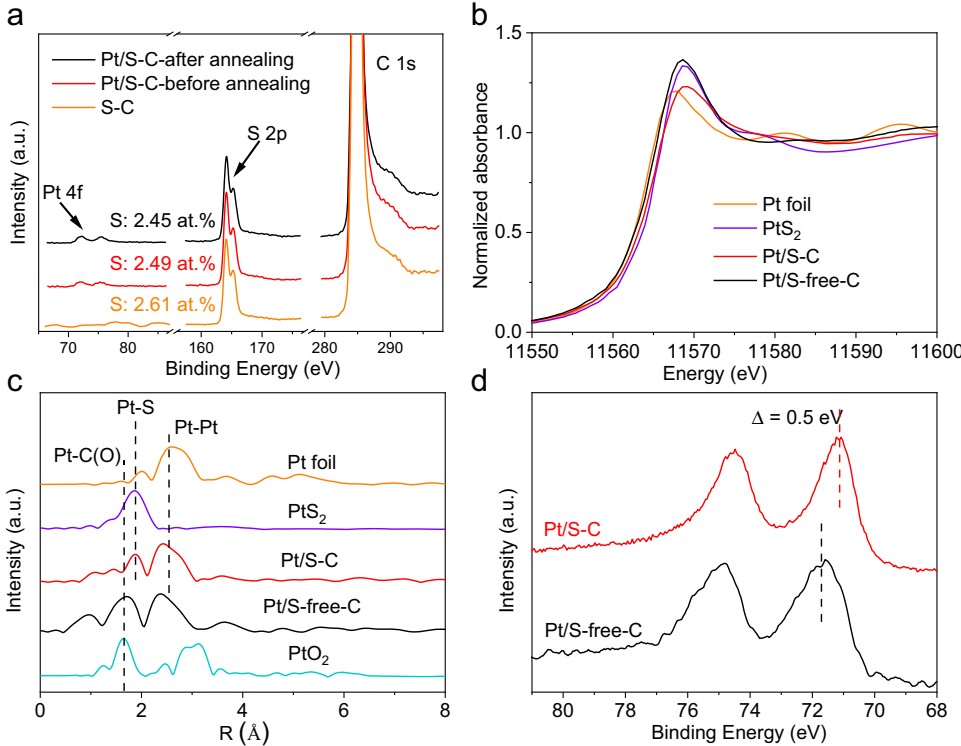

**Fig. 3 Spectroscopic characterizations. a** XPS survey spectra of the S–C support and the Pt/S–C catalyst before and after annealing treatment at 700 °C in 5% $H_2$/Ar for 600 min. **b** Pt $L_3$-edge XANES spectra of Pt/S–C, Pt/S-free-C, $PtS_2$, and Pt foil. **c** Fourier transform of the EXAFS spectra of Pt/S–C, Pt/S-free-C, $PtS_2$, $PtO_2$, and Pt foil. **d** In-situ XPS of Pt 4f on Pt/S–C and Pt/S-free-C after $H_2$ treatment at 200 °C for 30 min.

exhibited a considerably lower intensity of white line peak in the Pt $L_3$-edge XANES spectra than Pt/S-free-C (Fig. 3b), even though both of them had a similar particle size (Fig. 2a, e). The white line intensity directly reflects the unoccupied density of states of Pt 5d orbitals[37]. The lower intensity of white line peak for Pt/S–C suggests that the S–C support can donate electrons to Pt nanoclusters, which leads to an electron-enriched state of Pt nanoclusters on S–C. Such electronic interaction between Pt and S–C support has been confirmed previously by the density functional theory (DFT) calculations[38].

In the Fourier transforms of EXAFS data, Pt/S–C showed two prominent peaks at ~1.88 Å from the contributions of Pt binding with non-metallic element and ~2.5 Å from the Pt–Pt contributions, respectively (Fig. 3c). By comparing with the EXAFS of $PtS_2$ and $PtO_2$, we can safely conclude that the peak of 1.88 Å observed in Pt/S–C is exclusively associated to the Pt–S bonds, again implying the strong chemical interaction between Pt and S–C (Fig. 3c and Supplementary Fig. 18). Note that the Pt–Pt bonding distance in Pt/S–C is shortened than bulk platinum (~2.7 Å) owing to the existence of small Pt nanoclusters with low-lying electronic states in the range of 2.4–2.6 Å as revealed by theoretical studies[39]. Differently, the Pt/S-free-C catalyst showed the Pt–C (O) bonding at ~1.65 Å besides the Pt–Pt bonding at ~2.5 Å, without the Pt–S coordination environment anymore, confirming the "sulfur-free" feature of Pt/S-free-C.

To rule out the uncertainty of the oxidation states of Pt nanoclusters by air during ex-situ measurements, we further carried out in-situ XPS characterization to verify the electronic interactions between Pt and S–C. Therein, the as-prepared Pt/S–C and Pt/S-free-C catalysts were first treated with $H_2$ (1 bar) at 200 °C in a high pressure reactor, and then transferred to the analysis chamber for XPS measurements without exposing to air. We observed a significant red shift of the Pt 4f peak to a lower bind energy by 0.5 eV for Pt/S–C (71.1 eV) compared with

Pt/S-free-C (71.6 eV) (Fig. 3d), again confirming the electron donation from the S–C support to Pt nanoclusters.

In short, the above ex-situ XANES/EXAFS and in-situ XPS studies unambiguously certify that the sulfur atoms doped in the carbon matrix can form strong and thermally stable covalent bonds with Pt at the Pt/S–C interface, which plays a pivotal role in stabilizing metal nanoclusters at high temperatures.

**DFT calculations.** To elucidate the pivotal role of the interfacial metal-sulfur bonds in charge transfer (Supplementary Fig. 19) and suppressing metal nanocluster sintering (Fig. 4) at high temperatures, we carried out the DFT calculations. Based on the geometric details identified by the atom-resolution HAADF-STEM image (Supplementary Fig. 2), $Pt_{38}$/S-doped graphene (S–Graphene) model in the shape of truncated octahedron with exposed (111) and (200) planes was proposed for the first-principle examination with DFT. Considering that the degree of graphitization was much enhanced after the desulfurization treatment as determined by Raman spectrum analysis (Supplementary Fig. 20), a defect-free graphene layer was used as the substrate ($Pt_{38}$/graphene) for comparison for the DFT studies. The interfacial coordination of $Pt_{38}$/S-doped graphene is a Pt atom bonded to four S atoms, which is similar with the coordination shell of $PtS_2$ based on the EXAFS analysis[38].

To understand the direction of charge transfer between metal clusters and the S–C support, we calculated the valence band maximum (VBM) level of S–Graphene and the conduction band minimum (CBM) of $Pt_{38}$ cluster by using the Frontier Orbital Theory. Due to the VBM level of S–Graphene is higher than the CBM level of $Pt_{38}$ cluster, electrons will transfer from S–Graphene to $Pt_{38}$ cluster under the interaction of two orbitals, where the $Pt_{38}$ cluster spontaneously donate empty orbitals to capture electrons (Supplementary Fig. 19a). Moreover, we further

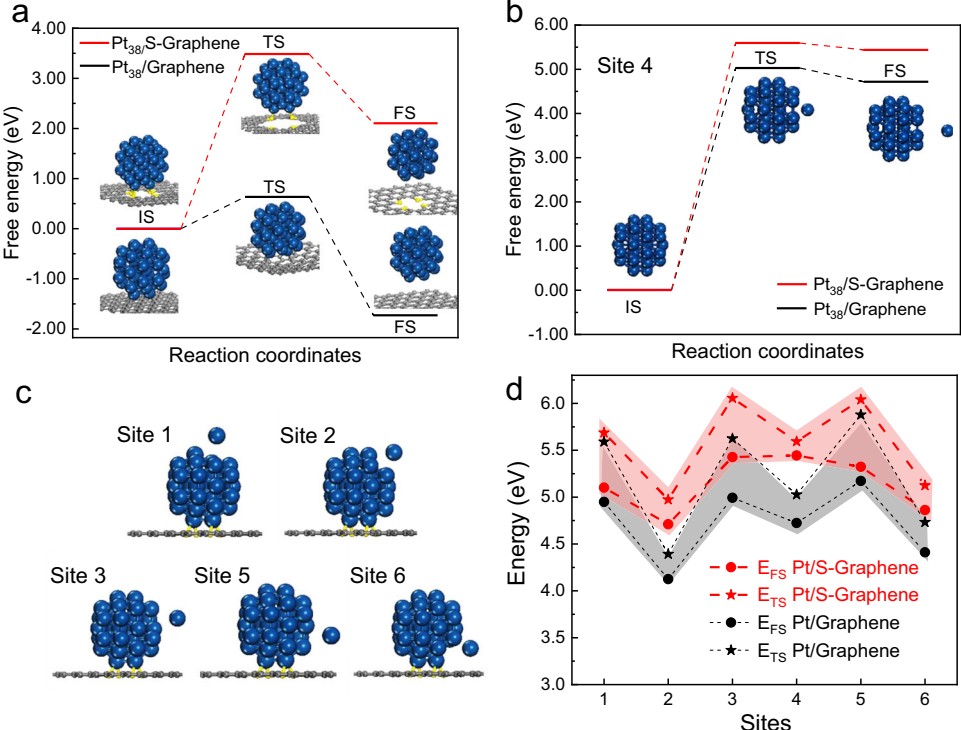

**Fig. 4 Theoretical investigations of nanoparticle diffusion and atom escaping. a** Energy barrier of $Pt_{38}$ cluster desorption on S–Graphene and Graphene. **b** Escape energy of one selected atom (site 4) from the $Pt_{38}$ cluster on S–Graphene and Graphene. **c** The selection of several other atoms at different positions for the escape energy calculations. **d** The escape energy of each atoms at different positions shown in (**c**). The higher energy barriers for both $Pt_{38}$ migration and individual atom escaping on the S–Graphene substrate support the experimentally observed sinter resistance of Pt nanoclusters on S–C.

explored the Bader charge analysis and found that $Pt_{38}$ cluster could capture 0.73 electrons from the S–Graphene (Supplementary Fig. 19b), which further demonstrated an electron-enriched state of Pt nanoclusters on S–C by the interfacial metal–sulfur electronic interactions.

For the simulation of sintering process, we considered both PMC and OR sintering paths. The PMC sintering mechanism involves the mobility of the whole particles on the support surface with Brownian motion collide[16]. In this case, the diffusion of particle needs to overcome the strength of adhesion at the metal/support interface. As shown in Fig. 4a, our DFT calculations suggested a much higher energy barrier of the $Pt_{38}$ nanocluster desorption on S–Graphene (3.49 eV) than that on Graphene (0.63 eV), indicating that the PMC process of Pt nanoclusters can be significantly suppressed on the S–C supports. The OR process involves the migration of individual atom driven by the differences of surface energy of nanoparticles, in which the atomic species are emitted from one nanoparticle, diffuse over the supports and attach to another nanoparticle, resulting in the gradual growth of larger particles and consumption of small particles[16]. By DFT simulation, we found that the escape energy of individual atom from $Pt_{38}$ cluster on S–Graphene was larger than that on Graphene (Fig. 4b–d), which also can be ascribed to the enhanced strength of interfacial adhesion at the $Pt_{38}$/S–Graphene owing to the Pt–S bonding[40]. The increased adhesion energy would greatly reduce the chemical potential of the surface metal atoms[17], which eventually suppress the inter-particle sintering kinetics by retarding the Ostwald ripening pathway. Overall, these calculation results clearly demonstrate that the strong metal/S–C interaction can dramatically enhance the anti-sintering ability by inhibiting both metal atom diffusion (OR path) and nanoparticle migration (PMC path).

**Catalytic propane dehydrogenation.** Taking into account the outstanding thermal stability of the S–C supported metal nanoclusters, we tested the catalysis application of Pt/S–C for the industrially important propylene production by PDH. It is well-documented that PDH is highly endothermic and thermo-dynamically limited and therefore requires a high reaction temperature, typically in the range of 400–600 °C[41]. Under such harsh conditions, it is extremely challenging to maintain a high propene selectivity and suppress the irreversible deactivation caused by undesirable metal nanoparticle sintering and heavy coking[11].

The PDH on Pt/S–C was first studied in a fixed-bed micro-reactor at 550 °C with 10 vol% $C_3H_8$ feeding. Pt/S-free-C, commercial Pt/C, Pt/$Al_2O_3$, and PtSn/$Al_2O_3$ catalysts were also tested under the same conditions. Notably, the Pt/S–C catalyst exhibited 40% conversion corresponding to a high initial propylene formation rate of 2028 mmol $g_{Pt}^{-1}$ $h^{-1}$ (Fig. 5a), which only slightly decreased to 1986 mmol $g_{Pt}^{-1}$ $h^{-1}$ (corresponding to 2.1% deactivation) after continuous operation for 600 min (Fig. 5b). The long-term durability of Pt/S–C for 1800 min operation further demonstrated its outstanding stability (Fig. 5c). In contrast, the propylene formation rate on the Pt/S-free-C catalyst decreased greatly decreased from 1745 to 1071 mmol $g_{Pt}^{-1}$ $h^{-1}$ (corresponding to 38.6% deactivation) after 600 min (Fig. 5a, b). Although Pt/$Al_2O_3$ exhibited a higher initial propylene formation rate (3303 mmol $g_{Pt}^{-1}$ $h^{-1}$) than Pt/S–C, the rate rapidly decayed to 1658 mmol $g_{Pt}^{-1}$ $h^{-1}$ after 120 min, which further decayed to 1340 mmol $g_{Pt}^{-1}$ $h^{-1}$ after 600 min (corresponding to 59.4% deactivation). The commercial Pt/C catalyst exhibited a much lower activity with initial rate of only 570 mmol $g_{Pt}^{-1}$ $h^{-1}$ as well as a poor stability with 73.5% deactivation (Fig. 5a, b). Besides, the conversion on PtSn/$Al_2O_3$

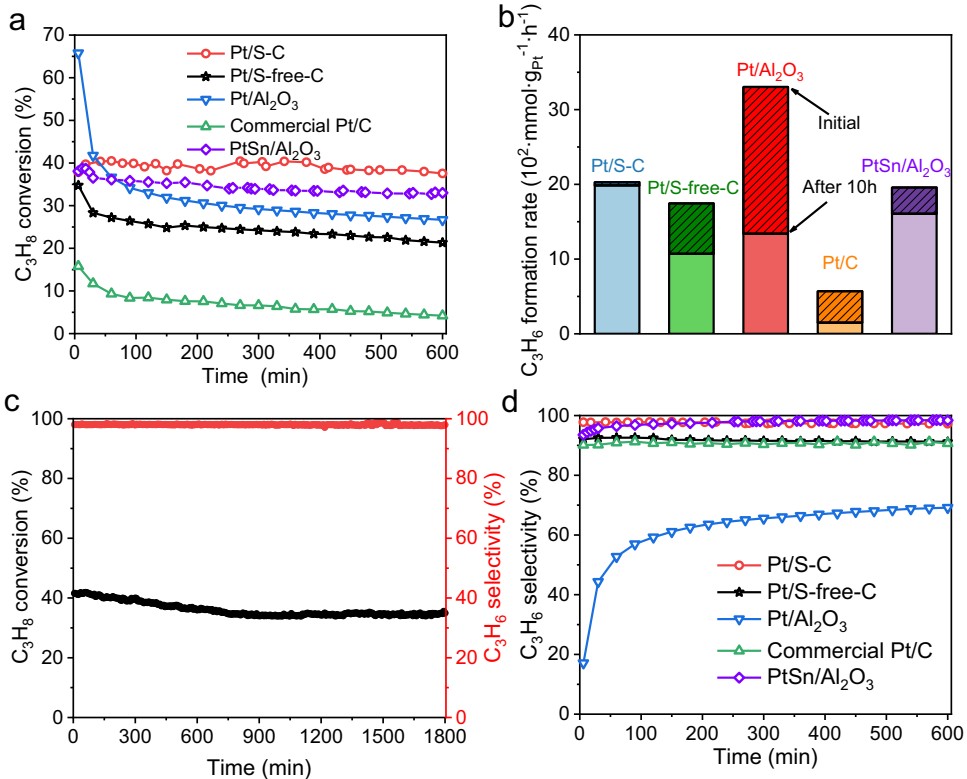

**Fig. 5 Catalytic performance for the high-temperature PDH. a** $C_3H_8$ conversion of Pt/S-C, Pt/S-free-C, commercial Pt/C, Pt/Al$_2$O$_3$ and PtSn/Al$_2$O$_3$ catalysts as a function of time during PDH. **b** Propylene formation rate at the initial stage and after 600 min continuous PDH operation. **c** Long-term stability test of Pt/S–C for PDH at 550 °C for 1800 min. **d** $C_3H_6$ selectivity of Pt/S-C, Pt/S-free-C, commercial Pt/C, Pt/Al$_2$O$_3$ and PtSn/Al$_2$O$_3$ catalysts as a function of time during PDH.

catalyst gradually decreased from 39 to 32% (corresponding to 17.9% deactivation) after 600 min (Fig. 5a), then further slowly decayed to 30% after 950 min (Supplementary Fig. 21). The first-order deactivation model was used to further analyze the catalysts stability. The much lower deactivation rate of 0.005 h$^{-1}$ for Pt/S–C quantitatively demonstrates its superior stability compared to Pt/S-free-C (0.05 h$^{-1}$), Pt/C (0.15 h$^{-1}$), Pt/Al$_2$O$_3$ (0.166 h$^{-1}$), and PtSn/Al$_2$O$_3$ (0.031 h$^{-1}$). Meanwhile, when the C$_3$H$_8$ concentration in feeding was increased from 10 vol% to 33 vol% and 50 vol%, although the conversion on Pt/S–C decreased from 40% to 19% and 13%, respectively, the Pt/S–C catalyst still showed a high stability with a comparatively lower deactivation rate of 0.003 h$^{-1}$ at 33 vol% C$_3$H$_8$ feeding and 0.005 h$^{-1}$ at 50 vol% C$_3$H$_8$ feeding. In contrast, for the Pt/S-free-C reference catalyst, the C$_3$H$_8$ conversion dramatically dropped from 30 to 18% with the increase of propane concentration (Supplementary Fig. 22). In addition, with the increases of weight hourly space velocity (WHSV) and temperature (Supplementary Fig. 23), the Pt/S–C catalyst could achieve a very high propane rate of 6645 mmol g$_{Pt}$$^{-1}$ h$^{-1}$.

Moreover, the Pt/S–C catalyst showed a high selectivity of ~98% in the whole catalysis process under 10% C$_3$H$_8$ concentration feeding (Fig. 5d), which is comparable to PtSn/Al$_2$O$_3$ and higher than the selectivity of Pt/S-free-C (~90%), Pt/C (90%), and Pt/Al$_2$O$_3$ (17–70%). The low selectivity of Pt/Al$_2$O$_3$ relative to carbon supported Pt catalysts is associated to the strong acidic sites of Al$_2$O$_3$, which promotes side reactions and coke formation[42]. Further, the Pt/Al$_2$O$_3$ could rapidly become deactivated through the blocking of the metal active site by coke deposition[43]. HAADF-STEM measurements was employed to monitor the morphology change of these catalysts before and after the PDH reaction. As speculated, serious sintering of the Pt

nanoclusters happened on Pt/S-free-C, Pt/C, Pt/Al$_2$O$_3$, and PtSn/Al$_2$O$_3$, while the size and morphology of the Pt nanoclusters in Pt/S–C were essentially unchanged (Supplementary Figs. 24, 25). Under high C$_3$H$_8$ concentration feedings of 33% and 50%, the selectivity of Pt/S-free-C showed a sharp decline from 90% to 80% and 50% and, therefore, exhibited rapid deactivation, while the selectivity of Pt/S-C showed slight decrease from 98% to 94% and 88% but still with a stable activity (Supplementary Fig. 26). When the reaction temperature rose to 600 °C, the Pt/S–C catalyst exhibited a slightly worse selectivity of around 92% than that under 550 °C with the decreased conversion from 55% to 40% after 1200 min, whereas the Pt/S-free-C catalyst showed a rapid deactivation at the beginning of the reaction (Supplementary Fig. 27). The widened performance gap between these two catalysts at more harsh conditions further demonstrated the pivotal role of the Pt–S interactions in promoting the selectivity and durability in catalyzing PDH.

We further compared the performance of the Pt/S–C catalyst with the reported monometallic Pt, Pt alloy, as well as metal oxide catalysts, in terms of the space time yield (STY) of propene formation, the selectivity to propene, and deactivation rate (Supplementary Fig. 28 and Table S3). At high C$_3$H$_8$ feed, the Pt/S–C catalyst exhibited a lower C$_3$H$_6$ STY than some metal oxides catalysts and most Pt alloy catalysts and acquired a higher C$_3$H$_6$ STY under high WHSV (5 h$^{-1}$) at 10 vol% C$_3$H$_8$ feed, which is superior to most monometallic Pt and even some Pt alloy catalysts. As for the selectivity, there is little difference between Pt alloy catalysts and the Pt/S–C (above ~90% for both kinds of catalysts), while some metal oxide and monometallic Pt catalysts showed low selectivity of 75–85%. In the term of deactivation rate, the Pt/S–C is obviously in the top position with outstanding stability and even better than most Pt-alloys catalysts.

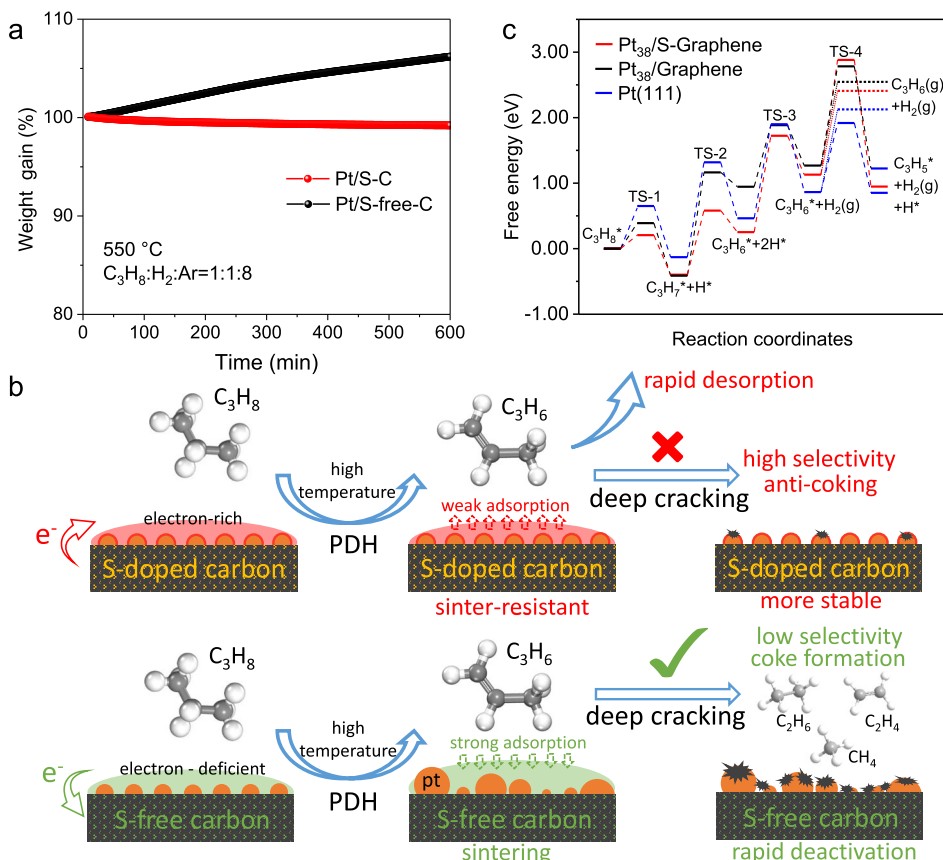

**Fig. 6 Experimental and theoretical interpretation of propylene selectivity. a** In-situ TGA of Pt/S–C and Pt/S-free-C under reactive atmosphere of PDH.
**b** Schematic illustration of the crucial role of interfacial electronic effects between Pt and S–C in promoting the selectivity and stability of PDH reaction.
**c** Calculated energy barriers for dehydrogenation steps of propane on $Pt_{38}$/S–Graphene, $Pt_{38}$/Graphene, and Pt (111). The dotted lines indicate the
desorption barriers of $C_3H_6$* to gaseous $C_3H_6$.

Considering the relatively low activity of Pt/S–C under 50 vol%
$C_3H_8$ feed, we further synthesized bimetallic Pt–Sn/S–C catalysts
with the target of further promoting $C_3H_6$ STY and selectivity
meanwhile maintaining low deactivation rate at high $C_3H_8$ feed
(Supplementary Fig. 29). The bimetallic Pt–Sn/S–C catalyst
exhibited an enhanced activity with a conversion of 25% and
also a promoted selectivity of 94% compared to monometallic
Pt/S–C catalysts (13% conversion and 88% selectivity) under the
identical reaction conditions. Meanwhile, the stability was well
maintained with a low deactivation rate constant of 0.003 h$^{-1}$ for
3000 min. These results make the Pt–Sn/S–C catalyst be one of
the best PDH catalysts (Supplementary Fig. 28 and Table S2). The
systematic optimization of the Pt–Sn/S–C catalyst and compre-
hensive screen of bimetallic combination of alloys may further
improve the performance, but it has seriously deviated from the
core concept of this work.

We also tried to apply the sulfur-doped TiO$_2$ support to
regeneration for PDH reaction under oxidation conditions. After
5 successive regeneration cycles, the performance remained little
change (Supplementary Fig. 30), indicating the feasibility of anti-
sintering under oxidation conditions of metal–S interaction on
industrially relevant metal oxide supports. The remarkable
sintering resistance of the Pt/S–C catalyst highlights its potential
technological value in realistic catalysis environments. For
example, for the water-gas shift (WGS) reaction, Pt/S–C also
exhibited a stable catalytic performance after continuous opera-
tion for 3500 min (Supplementary Fig. 31).

We carried out in-situ thermogravimetric analysis (TGA) under
reactive atmosphere of PDH to directly monitor the coke formation

at 550 °C (Fig. 6a). Encouragingly, the Pt/S–C exhibited outstanding
anti-coking ability with no any detectable coke accumulation under
the long-time reaction, indicating the easier propylene desorption
on electron-enriched Pt nanoclusters on the S–C support. In sharp
contrast, coke formation on the Pt/S-free-C catalyst was heavy,
reaching a 6.2% of weight gain after 600 min, which is ascribed to
the strong $C_3H_6$ adsorption and deep dehydrogenation on Pt/S-
free-C. The above XANES, EXAFS, and XPS have demonstrated
that the Pt nanoclusters were in an electron-enriched environment
induced by the electron donation from the S–C support to Pt. The
increased electron density of Pt nanoclusters can weaken the bond
between Pt and electron-rich group of C=C of propylene on
account of electrostatic repulsion[44], which would enhance the
propylene desorption and accordingly improve the PDH selectivity.
An additional advantage of the enhanced propylene desorption is to
efficiently avoid the further dehydrogenation of propylene and
eventually the coke formation (Fig. 6b), which is another
deactivation mechanism in addition to the metal particle
sintering[45].

## Discussion

We further performed the DFT calculations of the PDH reaction
paths on $Pt_{38}$/S–Graphene, $Pt_{38}$/Graphene, and Pt (111) surface to
understand the attribution of the strong Pt–S interaction to the
enhanced catalytic performance of Pt/S–C (Fig. 6c and Supple-
mentary Figs. 32–34). As revealed by previous theoretical studies,
the first dehydrogenation barrier ($C_3H_8$* → $C_3H_7$* + H*) is the
rate-determining step that reflects the PDH activity[46]. Our DFT

calculations suggested a much lower energy barrier of the first dehydrogenation step on $Pt_{38}$/S–Graphene (0.20 eV) than that on $Pt_{38}$/Graphene (0.41 eV) and Pt (111) (0.65 eV). Similarly, a relatively low dehydrogenation energy barrier on $Pt_{38}$/S–Graphene was also found in the second step ($C_3H_7^* \rightarrow C_3H_6^* + 2H^*$). These calculations results can explain well the much higher initial activity of Pt/S–C than that of Pt/S-free-C and Pt/C.

Subsequently, the $C_3H_6$ selectivity was evaluated by the competition between the $C_3H_6^*$ desorption and the deep C–H bond breaking[47]. On Pt (111), we determined the desorption energy of $C_3H_6$ to be 1.26 eV, which is higher the energy barrier of deep C–H bond breaking of $C_3H_6$ (1.05 eV), indicating that the deep $C_3H_6$ dehydrogenation is more favorable than the $C_3H_6$ desorption. Differently, on $Pt_{38}$/S–Graphene, the desorption energy barrier of $C_3H_6$ (1.27 eV) is significantly lower than the C–H bond breaking energy of $C_3H_6$ (1.75 eV), which would lead to a higher $C_3H_6$ selectivity over $Pt_{38}$/S–Graphene. Meanwhile, the smaller disparity between the $C_3H_6^*$ desorption and the deep C–H bond breaking on $Pt_{38}$/Graphene also prove the lower selectivity on Pt/S-free-C than that on Pt/S–C. The much increased activation barrier for deep dehydrogenation on $Pt_{38}$/S–Graphene could be associated to the electron-enriched state of $Pt_{38}$/S–Graphene induced by the electronic Pt–S interactions. The d-band of $Pt_{38}$/S–Graphene model shifts down away from Fermi level than $Pt_{38}$/Graphene from the change of density of states, which indicates the low activity for deep cracking on $Pt_{38}$/S–Graphene model. Further, $Pt_{38}$/S–Graphene transfers 0.08 e to $C_3H_6^*$ intermediate but $Pt_{38}$/Graphene only transfers 0.02 e, which may the cause of the down-shift of d-band and the decreased reactivity for deep dehydrogenation in $Pt_{38}$/S–Graphene model (Supplementary Fig. 35).

In summary, we have demonstrated that sulfur doped in carbon matrix can stabilize ~1 nm metal nanoclusters at high temperatures up to 700 °C in a reductive atmosphere. The strong chemical/electronic interactions between the metal nanoclusters and the S–C support significantly enhance the adhesion strength at the metal-support interface, which eventually suppresses the metal nanocluster sintering kinetics by retarding both the metal atom diffusion and the nanoparticle migration. For catalyzing the high temperature propane dehydrogenation reaction, the prepared Pt/S–C nanocluster catalysts with interfacial electronic effects exhibited distinctly better activity and selectivity than the state-of-the-art dehydrogenation catalysts, with only a slight performance deactivation. Our results show the feasibility of the sulfur-stabilization strategy to produce ~1 nm metal cluster catalysts for industrially relevant catalysis.

## Methods

**Materials and chemicals**. Silicon dioxide ($SiO_2$, 7 nm, S5130, 99%) were obtained from Sigma-Aldrich. 2,2'-bithiophene (98%) was purchased from J&K Scientific Ltd. All other chemicals were commercially available from Sinopharm Chemical Reagent Co. Ltd., China, including chloroplatinic acid ($H_2PtCl_6\cdot6H_2O$), ruthenium (III) chloride hydrate ($RuCl_3\cdot H_2O$), rhodium (III) chloride hydrate ($RhCl_3\cdot3H_2O$), osmium chloride ($OsCl_3$), iridium (III) chloride hydrate ($IrCl_3\cdot H_2O$), Tin (II) chloride dehydrate ($SnCl_2\cdot2H_2O$), titanium tetraisopropanolate, and thiourea. All the chemicals were used as received without further purification. Commercial Pt/$Al_2O_3$ (1 wt% Pt) and Pt/C (5 wt% Pt) were purchased from Alfa Aesar and Sigma-Aldrich, respectively. DI water (18.2 MΩ/cm) used in all experiments was prepared by passing through an ultra-pure purification system.

**Synthesis and sinter resistance of metal nanocluster catalysts**. The S–C supports were prepared by cobalt-assisted carbonization of molecular precursors (that is, 2,2'-bithiophene) with silica nanoparticles as templates, according to our previously reported works[30,31]. Noble metal cluster catalysts, including Pt, Ru, Rh, Os, and Ir were prepared with S–C as supports by a conventional impregnation method that involved the wetness impregnation of aqueous solution containing metal salt and subsequent thermal reduction in 5% $H_2$/Ar. Bimetallic (Pt–Ir and Ir–Ru) and multi-metallic (Pt–Ir–Ru) cluster catalysts were prepared with same method. Taking the synthesis of Pt/S–C for an example, $H_2PtCl_6$ was first mixed

with 100 mg S–C in a 250 mL round-bottom flask containing 100 mL DI water. After stirring overnight, the mixture was subjected to ultrasonic treatment for 1 h before drying by using a rotary evaporator. Finally, the dried powder was transferred to a tube furnace and thermally reduced at 700 °C under flowing 5% $H_2$/Ar for 2 h. The Pt loading was controlled to be 1 or 5 wt% and the actual Pt content in Pt/S–C catalyst was measured by inductively coupled plasma atomic emission spectrometry (ICP-AES). For the thermal stability study, the as-prepared catalysts were then annealed at 700 °C for 600 min in 5% $H_2$/Ar conditions.

The reference PtSn/$Al_2O_3$ catalysts was also prepared by wetness impregnation method. Briefly, 26.54 mg $H_2PtCl_6\cdot6H_2O$ and 17.35 mg $SnCl_2\cdot2H_2O$ were mixed with 1.0 g γ-$Al_2O_3$ support in a 250 mL round-bottom flask containing 100 mL DI water. After stirring overnight, the mixture was subjected to dry by using a rotary evaporator. This was followed by calcination in air at 500 °C for 2 h and then reduced in 5% $H_2$/Ar atmosphere at 600 °C for 2 h.

**Catalytic propane dehydrogenation**. The propane dehydrogenation reaction was performed in a fixed-bed quartz tube reactor at the atmospheric pressure and the products were analyzed by an online GC-14C gas chromatograph equipped with a flame ionization detector and a thermal conductivity detector. Before the dehydrogenation reaction, the catalysts were reduced in $H_2$/Ar at 200 °C for 1 h and then heated to 550 °C and fed with propane. Typically, the gas reactant contained 10 vol% propane and 10 vol% $H_2$ and a balance of Ar (total flow rate of 15 mL·min$^{-1}$). The operating temperature was 550 °C and the catalyst loading is 80 mg (1 wt% Pt), except for commercial Pt/C (20 mg, 5 wt% Pt). The weight hourly space velocity (WHSV) of propane was around 2 h$^{-1}$. The propane conversion and selectivity of the products were calculated from the following equations[48,49]:

Propane conversion:

$$\text{Conv.}\% = \frac{F_{\text{in}}(C_3H_8) - F_{\text{out}}(C_3H_8)}{F_{\text{in}}(C_3H_8)} \times 100\% \quad (1)$$

Products selectivity:

$$\text{Sel.}\% = \frac{\frac{n_i}{3}F_{\text{out}}(i)}{\sum \frac{n_i}{3}F_{\text{out}}(i)} \times 100\% \quad (2)$$

where $i$ represents hydrocarbon products in the effluent gas stream, $n_i$ is the number of carbon atoms of component $i$, and $F(i)$ is the corresponding flow rate.

$$\text{Carbon balance} = \frac{\frac{1}{3}[CH_4]_{\text{outlet}} + \frac{2}{3}[C_2H_4]_{\text{outlet}} + \frac{2}{3}[C_2H_6]_{\text{outlet}} + [C_3H_6]_{\text{outlet}} + [C_3H_8]_{\text{outlet}}}{[C_3H_8]_{\text{inlet}}}$$

(3)

Carbon balance typically ranged between 95% and 105% for all the reactions, which allows for ignoring the loss of carbon deposition to some extent.

The propene formation rate was determined under differential conditions. Each time node could correspond to the yield of propene at a current time. Then the propene formation rate and was calculated by the following formula that was defined as the moles of $C_3H_6$ formation per g $_{Pt}$ per hour.

$C_3H_6$ formation rate:

$$C_3H_6 \text{ formation rate} = \frac{F(C_3H_8) \times 60 \div 22.4 \times Y_{(C_3H_6)}}{m_{\text{cat}}w_{\text{Pt}}} \quad (4)$$

Attenuation rate of propylene formation:

$$\text{Attenuation rate} = \frac{C_3H_6\text{rate}_{\text{initial}} - C_3H_6\text{rate}_{\text{final}}}{C_3H_6\text{rate}_{\text{initial}}} \quad (5)$$

where $F(C_3H_8)$ represents the flow rate of propane, $Y$ and $m_{\text{cat}}$ are the yield of propylene and catalyst loading, respectively, and $W_{\text{Pt}}$ is the percentage of Pt weight loading in the catalyst.

Specific activity is defined as the moles of $C_3H_6$ formation per mole Pt atoms per second. A first-order deactivation model was used to evaluate the catalyst stability:

$$k_{\text{d}} = \frac{\ln\frac{1-X_{\text{final}}}{X_{\text{final}}} - \ln\frac{1-X_{\text{initial}}}{X_{\text{initial}}}}{t} \quad (6)$$

where $X_{\text{initial}}$ and $X_{\text{final}}$, respectively, represent the conversion measured at the start and the end of an experiment, and $t$ represents the reaction time (h), $k_{\text{d}}$ is the deactivation rate constant (h$^{-1}$). Higher $k_{\text{d}}$ values are indicative of rapid deactivation, that is, low stability.

**Catalytic water-gas shift reaction (WGS)**. The catalytic performance of the catalysts in the WGS reaction was evaluated in a fixed-bed flow reactor. The catalyst (10 mg) was pretreated by 5% $H_2$/Ar at 200 °C for 2 h. After that, the temperature was increased to 400 °C, and the catalyst was exposed to the WGS reaction mixture. The reactant gas consisted of 5% CO (flow rate: 30 mL min$^{-1}$) and water vapor at 46 °C (water vapor pressure: 10.094 kPa) balanced with Ar that yielded the $P_{CO}/P_{H2O}$ ratio of 1:2. All catalysts were heated to the desired reaction temperatures at a rate of 1 K min$^{-1}$, and the steady state compositions of the effluent gas were analyzed with an online gas chromatograph (FULI 9790II) with a TCD attached to a TDX column. The catalytic activity was calculated by the change

in the CO concentrations of the inlet and outlet gases. The WGS rate was calculated based on the total Pt content.

**DFT Calculations**. DFT calculations were performed by using the Vienna Ab-initio Simulation Package (VASP) with the projector augmented wave method for the core region and a plane-wave kinetic energy cutoff of 420 eV. The generalized gradient approximation method with Perdew-Burke-Ernzerh of (PBE) functional for the exchange-correlation term was used. The convergence of energy and forces were set to be less than $1 \times 10^{-5}$ eV and 0.05 eV/Å, respectively. In order to study the thermal stability and catalytic activity of $Pt_{38}$/S–Graphene and $Pt_{38}$/Graphene, we built models of S-doped graphene layer and graphene layer supported $Pt_{38}$ cluster. Supercells of $8 \times 8$ graphene unit cells were used to simulate situations in experiments, respectively. A set of $3 \times 3 \times 1$ $k$-points were sampled by using gamma-centered Monkhorst-Pack scheme to describe the Brillouin zone. After geometry optimizations, neb computations were established to find transition states in simulations of $Pt_{38}$ cluster separation from S–Graphene and Graphene. We artificially pull $Pt_{38}$ cluster separation away from substrate to calculate reaction path. A set of $1 \times 1 \times 1$ $k$-points were sampled by using gamma-centered Monkhorst-Pack scheme to describe the Brillouin zone. Energy barriers of the $Pt_{38}$ nanocluster diffusion on S–Graphene/Graphene were calculated as follows:

$$E_{barrier} = E_{trans} - E_0 \qquad (7)$$

Where $E_{trans}$ represent the energy of transition state which only one vibration frequency is a virtual frequency, $E_0$ represent the energy of structures after geometry in the previous step that can considered as ground state structure. The escape energy of individual atom from $Pt_{38}$ cluster can also be calculated by the above method.

As for catalytic activity, a series of complex calculations were carried out. At first, we chose a suitable reaction path to simulate the catalytic reaction of propane[46,49]. Five points in the reaction path were selected to describe the intermediates in whole reaction process. Similarly, neb computations were established to obtain energy barriers. A set of $1 \times 1 \times 1$ $k$-points were sampled by using gamma-centered Monkhorst-Pack scheme to describe the Brillouin zone. Energy barriers of the whole process can be given by following equation:

$$E_{barrier\_N} = E_{trans\_N} - E_N \qquad (8)$$

Where $E_{trans\_N}$ represent the energy of transition state from point (N) to point (N+1), $E_N$ represent the energy of point (N). N values range from 1 to 5.

**In-situ TGA**. In-situ TGA measurements were performed on a TGA Q600 (TA Instruments). Typically, 10 mg of sample was reduced at 200 °C in 10% $H_2$/Ar (flow rate 20 mL/min) for 60 min. Then, the feed gas, consisting with 10% propane and 10% $H_2$ with Ar as the balance gas, was introduced to the reactor for 10 h. The evolution of sample weight during the coke deposition was then recorded.

**XAFS experiments and data analyses**. XAFS spectra at Pt $L_3$-edge were obtained on the 1W1B beamline of Beijing Synchrotron Radiation Facility (BSRF) operated at 2.5 GeV and 250 mA and on the BL14W1 beam line of Shanghai Synchrotron Radiation Facility (SSRF) operated at 3.5 GeV and 220 mA. The raw data analyses were conducted by using Athena program in IFEFFIT software package. The energy was first calibrated, then the pre-edge background of spectrum was subtracted and post-edge was normalized. $K^3$-weighted EXAFS oscillations ranging from 2.5 to 12.2 Å were Fourier transformed to obtain a radial distribution function. The data fitting was carried out using Artemis program in IFEFFIT.

**In-situ XPS measurements**. In-situ XPS were conducted at the photoemission end-station at the beamline BL10B of the National Synchrotron Radiation Laboratory (NSRL) in Hefei, China. The end-station is composed of an analysis chamber, a preparation chamber, a load-lock chamber, and a high pressure reactor. The analysis chamber with a base pressure of $<2 \times 10^{-10}$ torr is connected to the beamline with a VG Scienta R3000 electron energy analyser and a twin-anode X-ray source. The Pt/S–C and Pt/S-free-C catalysts were treated by $H_2$ (1 bar) at 200 °C for 30 min in the high pressure reactor. Afterward, the reactor was pumped with the pressure down to $<10^{-8}$ torr, followed by transferring them to the analysis chamber for XPS measurement without exposing to air.

**Characterization**. Low-magnification HAADF-STEM images were obtained on FEI Talos F200X operated at 200 kV. Atomic resolution HAADF-STEM images were obtained on probe aberration-corrected JEM ARM200F (S) TEM operated at 200 kV. Energy Dispersive Spectroscopy (EDS) mappings and line scanning were carried out on FEI Talos F200X, equipped with Super X-EDS system (four systematically arranged windowless silicon drift detectors) at 200 kV. ICP-AES measurements were performed using an Atom scan Advantage Spectrometer (Thermo Ash Jarrell Corporation). XPS measurements were conducted on a VG ESCALAB MK II X-ray photoelectron spectrometer with an exciting source of Mg Kα = 1253.6 eV. $N_2$ sorption analysis was conducted using an ASAP 2020 accelerated surface area and porosimetry instrument (Micromeritics), equipped with automated surface area, at 77 K using BET calculations for the surface area. The pore

size distribution plot was analyzed from the adsorption branch of the isotherm based on the quenched-solid density functional theory (QSDFT).

## Data availability

The data that support the findings of this study are available from the corresponding author upon request.

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

## Acknowledgements

This work was supported by the National Key Research and Development Program of China (Grant 2018YFA0702001, 2018YFA0208603, 2016YFA0200602), the National Natural Science Foundation of China (Grant 21671184, 22071225, 22073087, and 21890751), the Fundamental Research Funds for the Central Universities (Grant WK2060190103), the Joint Funds from Hefei National Synchrotron Radiation Laboratory (Grant KY2060000175), and the Recruitment Program of Thousand Youth Talents. We acknowledge Beijing Synchrotron Radiation Facility (1W1B beam line), Hefei National Synchrotron Radiation Laboratory (BL10B beam line), and Shanghai Synchrotron Radiation Facility (BL14W1 beam line) for the synchrotron beam time.

## Author contributions

H.-W.L. and P.Y. conceived the research. P.Y. performed the catalyst synthesis and characterizations. X.-J.W. and X.L. performed the DFT calculation. J.-L.L., Y.-F.M., and S.C. performed the catalytic tests. S.-Q.C. performed the X-ray absorption spectra measurements. P.Y. and X.-S Z. performed the in-situ XPS measurements. P.Y., X.L., H.-W.L., X.-J.W., and J.-L.L. co-wrote the paper. All authors discussed the results and commented on the manuscript.

## Competing interests

The authors declare no competing interests.
