## [Peer Review File · Nature Communications]

REVIEWER COMMENTS

Reviewer #1 (Remarks to the Author):

In their manuscript Yin et al. reported about the effect of sulfur on stability of metal nanoclusters against their sintering. The authors have shown that metal nanoclusters (Pt, Ru, Rh, Os, or Ir) supported on sulfur-doped carbon support exhibit remarkable sinter-resistant properties unlike their counterparts supported on the undoped support. It has been concluded that the enhanced stability of the former arises from strong metal-sulfur interaction. This conclusion is not novel (see comment 1). Practical relevance of Pt/S-C catalyst was also not justified (see comments 2-5). Specific comments are given below.

1. The novelty of the idea of stabilizing metal nanoparticles against sintering by interaction with sulfur on carbon support is doubtful. Several papers (J. Mater. Chem., 2009, 19, 5934–5939; J. Mater. Chem. A, 2017, 5, 19467-19475; Scientific Reports, 2019, 9, 12704) have shown that sulfur-doped carbon materials are promising supports for Pt-based catalysts. The addition of sulfur into support was shown to improve thermal stability and dispersion of Pt particles due to the strong metal-support interaction between the sulfur and Pt atoms. Although there is an obvious similarity between the current work and the works listed above, the present authors did not refer to previous studies.

2. It is well-known that unpromoted supported Pt species tend to sinter under propane dehydrogenation (PDH) conditions. Therefore, a typical industrial catalyst contains Sn as a promoter for stabilizing Pt particles. In order to demonstrate the practical potential of Pt/S-C in propane dehydrogenation the authors should compare their catalyst with Sn-Pt/Al₂O₃ under the same reaction conditions.

3. Taking into account the difference in experimental conditions in the present and previous studies, any correct comparison of the performance of Pt/S-C and the state-of-the-art Pt-based catalysts (Supplementary Table 3) is not possible. Moreover, unlike the state-of-the-art catalysts listed in Table 3, Pt/S-C was tested using a quite diluted mixture (only 10 vol% C₃H₈) that may be a reason for the high selectivity and low deactivation constant. Moreover, the present authors do not provide degrees of propane conversion. In general, catalysts used in the PDH reaction can be compared at same propane conversion obtained at same temperature and using same reaction feed. Thus, the superior performance of Pt/S-C is not justified.

4. It is also not clear how the authors determined the rate of propene formation. Was it determined in a differential reactor? What was the degree of propane conversion?

5. Another serious concern about the applicability of the developed catalysts for propane dehydrogenation is the absence of any data related to anti sintering performance under air conditions. Do Pt species sinter in air? This is actually a problem of commercial Pt-containing catalysts.

Reviewer #2 (Remarks to the Author):

This manuscript discusses a new method of stabilizing small metal nanoclusters for heterogeneous

catalysis using sulfur-doped carbon matrix support. This approach allows high thermal stability of 1 nm metal nanoclusters of Pt, Ru, Rh, Os, and Ir at elevated temperatures up to 700 °C, preventing thermal sintering and consequent deactivation. The S-C bond in the matrix enhanced the adhesion strength between the nanoclusters and the support, which arises from the interfacial metal-sulfur interaction that retards metal atom diffusion and ripening. A highly efficient and stable catalyst for propane dehydrogenation was developed based on sulfur-doped carbon-supported Pt nanocluster with 98% selectivity and extended durability for 1800 minutes. The work presented in the manuscript is scientifically sound and presented concisely and clearly. This work provides new synthetic approaches for heterogeneous catalysis by solving an important issue of metal sintering and deactivation during high-temperature catalysis. I therefore recommend its publication after addressing the following issues:

1. The authors need to point out the novelty and advantages of their stabilizing method more clearly. Hu et al. were able to demonstrate that by using the high-temperature shockwave method, they also could stabilize small metal nanoclusters on different supports such as carbon, C₃N₄, and TiO₂ through metal-defects bonds, as the authors themselves mentioned in the manuscript (Yao, Y. et al. High-temperature shockwave stabilized single atoms. *Nat. Nanotechnol.* 14, 851-857). Then how could the authors distinguish their work from Hu's work? What are the main advantages of the sulfur-carbon support approach comparing to other methods? These need to be addressed clearly in the manuscripts.
2. The authors mentioned that "The lower intensity of white line peak for Pt/S-C suggests that the S-C support can donate electrons to Pt nanoclusters, which leads to an electron-enriched state of Pt nanoclusters on S-C. Such electronic interaction between Pt and S-C support has been confirmed previously by the density functional theory (DFT) calculations." Can you explain further how this electron donation from S-C support to Pt happens? What is the primary mechanism behind it? How Pt accepts these electrons and in which state?
3. What is the primary function of carbon here? Can this sulfur stabilizing approach work with other supports that do not contain carbon? Can it work, for example, with TiO₂ as support?
4. I suggest the authors change Fig. 1 to a more demonstrative schematic illustration. The current figure is a bit confusing and does not give precise overall info of the manuscript story.
5. The authors also need to draw another figure representing the overall mechanism of S-C stabilizing with metal nanocluster interaction, especially demonstrating the following statement "The increased electron density of Pt nanoclusters can weaken the bond between Pt and an electron-rich group of C=C of propylene on account of electrostatic repulsion⁴¹, which would enhance the propylene desorption and accordingly improve the PDH selectivity." This figure could also include the overall dehydrogenation process of propane using Pt-supported S-C catalyst.
6. The authors have focused on using only single metal nanoclusters in the catalysis, but what about bimetallic nanoclusters? Such as Pt-Ir/S-C catalysts? Or even tertiary nanoclusters? Can they also work in this stabilizing method?
7. I also suggest that the authors demonstrate the applicability of their catalysis in other high-temperature reactions, such as methane or CO conversion.

Reviewer #3 (Remarks to the Author):

Yin et al. reported a sulfur doped carbon matrix to stabilize the various metal nanoclusters (Pt, Ru, Rh, Os, and Ir). Among the synthesized nanoclusters, Pt was found to perform high selectivity in propane dehydrogenation. Despite the topic may be interesting to the community, several uncertainties are required to be included in order to draw a more rigorous scientific conclusion for the current experimental findings. Therefore, the current form is suggested to undergo a major revision subject to further review with the additional improvement.

1. Figure 3(c) contains the Fourier transform of the EXAFS spectra of Pt/S-C, Pt/S-free-C, PtS₂, PtO₂, and Pt foil. Both PtS₂ and PtO₂ are bulk materials, however, the second coordination feature of PtS₂ is absent in the corresponding measurement. That is consequently challenges the accuracy of Pt-S bond identity in the Pt/S-C sample.
2. Additionally, the comparable Fourier transform of the EXAFS results were not provided for the Ru, Rh, Os, and Ir cases. A consistent comparison showing the superior selectivity of Pt/S-C over other tested metal clusters was not on the same basis.
3. The selection of M₃₈ cluster model to represented the synthesize M/S-C catalysts need more supportive evidences. The selection of size and shape of DFT models show any consistent evidence with any experimental observables?
4. The DFT results of Figure 4 are also confusing. Figure 4(a) should be renamed as Pt₃₈ cluster desorption since the FS geometry of Pt₃₈/S-C is substantially away from the graphene plane. The individual atom "escape energy" in Figure 4(b) is ambiguous due to the selection of the "single atom" is not well-defined, and the "escape energy" is believed to be substantially subject to the atom selection.
5. The *C₃H₆ intermediate of Fig 5(f) show significant stability with the Pt₃₈/S-C model, and such an enhanced stability could be highly related to the observed dehydrogenation selectivity. More insightful electronic structure analysis should have been provided to show absorbed geometric effect, charged transfer effect as well as the intrinsic bonding property of metal!

Reviewer #1 (Remarks to the Author):

In their manuscript Yin et al. reported about the effect of sulfur on stability of metal nanoclusters against their sintering. The authors have shown that metal nanoclusters (Pt, Ru, Rh, Os, or Ir) supported on sulfur-doped carbon support exhibit remarkable sinter-resistant properties unlike their counterparts supported on the undoped support. It has been concluded that the enhanced stability of the former arises from strong metal-sulfur interaction. This conclusion is not novel (see comment 1). Practical relevance of Pt/S-C catalyst was also not justified (see comments 2-5). Specific comments are given below.

1. The novelty of the idea of stabilizing metal nanoparticles against sintering by interaction with sulfur on carbon support is doubtful. Several papers (J. Mater. Chem., 2009, 19, 5934–5939; J. Mater. Chem. A, 2017, 5, 19467-19475; Scientific Reports, 2019, 9, 12704) have shown that sulfur-doped carbon materials are promising supports for Pt-based catalysts. The addition of sulfur into support was shown to improve thermal stability and dispersion of Pt particles due to the strong metal-support interaction between the sulfur and Pt atoms. Although there is an obvious similarity between the current work and the works listed above, the present authors did not refer to previous studies.

Response: We agree with the reviewer that the above mentioned literatures have reported sulfur-doped carbon materials as promising supports for Pt-based catalysts, which were not cited in the original manuscript. **Actually we cited another two papers that discussed the stabilization of metal clusters based on the interactions between metal and nitrogen or sulfur atoms (refs. 32 and 33).** Here we would like to explain why we cited refs. 32 and 33 but not the mentioned literatures in the original manuscript.

In all the mentioned literatures, **the S-doped carbon supports were reported to stabilize Pt nanoparticles of 3~5 nm, instead of Pt nanoclusters.** For example, in the first paper (J. Mater. Chem., 2009, 19, 5934–5939), the authors demonstrated that Pt nanoparticles supported on sulfur-containing ordered mesoporous carbon (S-OMC)

exhibited thermal stability at 600 °C under N₂ flow and that the sizes of Pt nanoparticles increased from 3.14 nm to 4.50 nm. Another two papers (J. Mater. Chem. A, 2017, 5, 19467-19475; Scientific Reports, 2019, 9, 12704) studied the electrochemical stability (instead of thermal stability) of about 3 nm Pt nanoparticles on sulfur-doped carbon.

In contrast, **our current work focuses on the thermal stability of metal nanoclusters of around 1 nm rather than nanoparticles of 3~5 nm**, as we highlighted in the Title (*Sulfur stabilizing metal nanoclusters on carbon at high temperatures*), Abstract, and Introduction parts. The motivation of our work is that **metal nanoclusters have some distinct features compared to nanoparticles but are extremely difficult to stabilize even on high-surface-area supports, owing to the remarkably improved surface free energy**. This is particularly challenging for metal nanocluster catalysts in the applications of high-temperature reactions, where thermal sintering induced catalyst deactivation is often the key issue.

Therefore, **in the original manuscript, we cited two papers that discussed the stabilization of metal clusters based on the interactions between metal and nitrogen (ref. 32) and sulfur atoms (ref. 33)**.

Further, although some doped carbon supports (including the literatures mentioned by the reviewer) have been reported to stabilize Pt nanoparticles of 3~5 nm up to 600 °C, to our knowledge, **so far no one has reported that metal clusters of 1 nm could be stabilized on carbon supports up to 700 °C**. We thus believe that these published works on the nanoparticles do not weaken the novelty of our work on metal nanoclusters and that our work represents a solid step toward the applications of metal nanocluster catalysts under realistic technical conditions.

According to the reviewer's comments, we also cited the mentioned three papers in the revised manuscript to better clarify the background as below:

Page 4: *“Although some sulfur-doped carbon supports have been reported to stabilize Pt nanoparticles of 3~5 nm up to 600 °C, to our knowledge, so far no one has reported that metal clusters of 1 nm could be stabilized on carbon supports up to 700 °C.”*

2. It is well-known that unpromoted supported Pt species tend to sinter under propane dehydrogenation (PDH) conditions. Therefore, a typical industrial catalyst contains Sn as a promoter for stabilizing Pt particles. In order to demonstrate the practical potential of Pt/S-C in propane dehydrogenation the authors should compare their catalyst with Sn-Pt/Al₂O₃ under the same reaction conditions.

Response: Many thanks for the reviewer's valuable comment here. We have supplemented the PtSn/Al₂O₃ catalysis data in the revised manuscript. The commercial γ -Al₂O₃ support was purchased from Alfa-Aesar. The PtSn/Al₂O₃ catalyst was prepared by the conventional impregnation method. The prepared PtSn/Al₂O₃ was then tested for the propane dehydrogenation under the same reaction conditions (Fig. R1). The selectivity of PtSn/Al₂O₃ (98%) was comparable to Pt/S-C, but the conversion on PtSn/Al₂O₃ catalyst gradually decreased from 39% to 32% (corresponding to 17.9% deactivation) after 600 min, then further slowly decayed to 30% after 950 min. The deactivation rate of PtSn/Al₂O₃ was 0.031 h⁻¹ within 600 min, which is also much higher than that of Pt/S-C (0.005 h⁻¹). Meanwhile, sintering of PtSn nanoclusters happened by monitoring the morphology change of PtSn/Al₂O₃ catalysts before and after the PDH reaction (Fig. R2), which results in the attenuation of activity despite of its high selectivity.

The above results and discussion on the PtSn/Al₂O₃ catalyst have been added in the revised manuscript (page 8, Fig. 5) and Supplementary Information (Figs. S21 and S25).

Figure R1. (a) C₃H₈ conversion and (b) C₃H₆ selectivity of Pt/SC, Pt/S-free-C, commercial Pt/C, Pt/Al₂O₃, and PtSn/Al₂O₃ catalysts under the same PDH reaction

conditions. (c) Long-term stability test of PtSn/Al₂O₃ for PDH at 550 °C for 950 min.

Figure R2. HAADF-STEM images of fresh (a) and spent (b) of PtSn/Al₂O₃. The spent catalysts were used for PDH at 550 °C for 950 min.

3. Taking into account the difference in experimental conditions in the present and previous studies, any correct comparison of the performance of Pt/S-C and the state-of-the-art Pt-based catalysts (Supplementary Table 3) is not possible. Moreover, unlike the state-of-the-art catalysts listed in Table 3, Pt/S-C was tested using a quite diluted mixture (only 10 vol% C₃H₈) that may be a reason for the high selectivity and low deactivation constant. Moreover, the present authors do not provide degrees of propane conversion. In general, catalysts used in the PDH reaction can be compared at same propane conversion obtained at same temperature and using same reaction feed. Thus, the superior performance of Pt/S-C is not justified.

Response: Thanks for the reviewer's valuable comments here. As suggested by the reviewer, it is impossible to make a more fair performance comparison between different catalysts due to the various experimental conditions. We quite agree with the reviewer on this issue and have deleted the comparison table and relevant discussion from the revised manuscript to make the discussion more objective. Meanwhile, we further supplemented the catalytic experiments of high-concentration C₃H₈ with the ratio of propane of 33 vol% and 50 vol% in feeding (Fig. R3). As a result, when the C₃H₈ concentration was increased from 10 vol% to 33 vol% and 50 vol%, the conversion of Pt/S-C decreased from 40% to 20% and 13%, respectively. Despite of a

slight decrease in selectivity (from 98% to 95%) at 33 vol% C₃H₈ feeding, the Pt/S-C showed a high stability in activity with a comparatively lower deactivation rate of 0.003 h⁻¹, which clearly demonstrates the superior stability of Pt/S-C. The selectivity dropped to around 88% at 50 vol% C₃H₈ feeding, but the Pt/S-C still showed a stable activity. In contrast, for the Pt/S-free-C reference catalyst, the propane conversion dramatically dropped from 30% to 18% after 1000 min and the selectivity also showed a sharp decline from 80% to 50% with the increase of propane concentration.

The above results and discussion on the catalytic experiments of high-concentration C₃H₈ have been added in the revised manuscript (page 8) and Supplementary Information (Fig. S22).

Figure R3. (a-b) Catalytic performance of Pt/S-C (a) and Pt/S-free-C (b) under 33 vol% C₃H₈ (C₃H₈: H₂: Ar =1:1:1). (c-d) Catalytic performance of Pt/S-C (c) and Pt/S-free-C (d) under 50 vol% C₃H₈ (C₃H₈: H₂=1:1, no Ar dilution).

4. It is also not clear how the authors determined the rate of propene formation. Was it determined in a differential reactor? What was the degree of propane conversion?

Response: Thanks for the reviewer's valuable comments here. According to the reviewer's comments, we have changed the ordinate to the propane conversion in revised manuscript. As described in the original manuscript (Page 13), the propane dehydrogenation reaction was performed in a fixed-bed quartz tube reactor at the atmospheric pressure and the products were analyzed by an online GC-14C gas chromatograph equipped with a flame ionization detector and a thermal conductivity detector. The propene formation was defined as the moles of C₃H₆ formation per g Pt per hour from the following formula:

$$C_3H_6 \text{ formation rate} = \frac{F(C_3H_8) \times 60 \div 22.4 \times Y_{(C_3H_6)}}{m_{cat}W_{Pt}}$$

where F (C₃H₈) represents the flow rate of propane, Y_(C₃H₆) and m_{cat} are the yield of propylene and the weight of catalysts, respectively, and W_{pt} is the percentage of Pt weight loading in the catalyst.

5. Another serious concern about the applicability of the developed catalysts for propane dehydrogenation is the absence of any data related to anti sintering performance under air conditions. Do Pt species sinter in air? This is actually a problem of commercial Pt-containing catalysts.

Response: Many thanks for the reviewer's comments on this issue. To be frank, as a carbon support, our S-C material would burn out at high temperature under oxygen atmosphere. Therefore, we cannot carry out the sintering experiments under air conditions for the carbon supported metal catalysts. The carbon supported metal catalysts are thus only applicable to the high-temperature reactions under reductive atmospheres, for example, the propane dehydrogenation reaction demonstrated here, the hydrogenation of CO₂ (Ind. Eng. Chem. Res. 2020, 59, 35, 15393–15423), semi-hydrogenation of acetylene (Nat. Commun. 2020, 11, 3324) or methane anaerobic conversion (Nat. Nanotechnol. 2019, 14, 851-857).

Part of above discussion has been added in the revised manuscript (page 4).

Reviewer #2 (Remarks to the Author):

This manuscript discusses a new method of stabilizing small metal nanoclusters for heterogeneous catalysis using sulfur-doped carbon matrix support. This approach allows high thermal stability of 1 nm metal nanoclusters of Pt, Ru, Rh, Os, and Ir at elevated temperatures up to 700 °C, preventing thermal sintering and consequent deactivation. The S-C bond in the matrix enhanced the adhesion strength between the nanoclusters and the support, which arises from the interfacial metal-sulfur interaction that retards metal atom diffusion and ripening. A highly efficient and stable catalyst for propane dehydrogenation was developed based on sulfur-doped carbon-supported Pt nanocluster with 98% selectivity and extended durability for 1800 minutes. The work presented in the manuscript is scientifically sound and presented concisely and clearly. This work provides new synthetic approaches for heterogeneous catalysis by solving an important issue of metal sintering and deactivation during high-temperature catalysis. I therefore recommend its publication after addressing the following issues:

1. The authors need to point out the novelty and advantages of their stabilizing method more clearly. Hu et al. were able to demonstrate that by using the high-temperature shockwave method, they also could stabilize small metal nanoclusters on different supports such as carbon, C₃N₄, and TiO₂ through metal-defects bonds, as the authors themselves mentioned in the manuscript (Yao, Y. et al. High-temperature shockwave stabilized single atoms. Nat. Nanotechnol. 14, 851-857). Then how could the authors distinguish their work from Hu's work? What are the main advantages of the sulfur-carbon support approach comparing to other methods? These need to be addressed clearly in the manuscripts.

Response: Thanks for the reviewer's valuable comments here. Hu et al. creatively used controllable high-temperature shockwaves to synthesize and stabilize single atoms by forming thermodynamically favorable metal-defect bonds. The reported shockwave method is facile and universal. Unlike metal-defect bonds, the key of our work is to construct strong metal-sulfur bonds to enhance the adhesion strength at the metal-support interface and eventually suppress the metal atom diffusion and the nanoparticle

migration, which would distinguish our work from Hu's work. In addition to having the superior stability and similar universality for various metals, our work is still amenable to the large-scale manufacture and high metal loading catalysts. More important, different from the traditional presumption that sulfur is a poison reagent for metal, the metal-sulfur interface we proposed endow the catalyst with unique electronic interactions, which are often propitious to high activity and selectivity (this work for PDH reaction).

As suggested by the reviewer, we have reinforced the relevant discussion to make the novelty and advantages of our methods more clear.

2. The authors mentioned that "The lower intensity of white line peak for Pt/S-C suggests that the S-C support can donate electrons to Pt nanoclusters, which leads to an electron-enriched state of Pt nanoclusters on S-C. Such electronic interaction between Pt and S-C support has been confirmed previously by the density functional theory (DFT) calculations." Can you explain further how this electron donation from S-C support to Pt happens? What is the primary mechanism behind it? How Pt accepts these electrons and in which state?

Response: Many thanks for the reviewer's comments on this issue. We tried to make further explanation about the charge transfer by using the Frontier Orbital Theory (Fig. R4). We calculated the valence band maximum (VBM) level of S-Graphene and the conduction band minimum (CBM) of Pt₃₈ cluster. Due to the VBM level of S-Graphene is higher than the CBM level of Pt₃₈ cluster, electrons will transfer from S-Graphene to Pt₃₈ cluster under the interaction of two orbitals, where the Pt₃₈ cluster spontaneously donate empty orbitals to capture electrons. Moreover, according to the reviewer's comments, we explored the Bader charge analysis and found that the Pt₃₈ cluster could capture 0.73 electrons from the S-Graphene. Notably, not all of Pt atoms have obtained electrons with negative valence, where we consider sulfur has induced dipole changes

of Pt₃₈ cluster. The state of each Pt atom was shown in Fig. R4c.

The above simulation results and related discussion have been added in the revised manuscript (page 7) and Supplementary Information (Fig. S19).

Figure R4. (a) Illustration of the charge transfer direction based on the Frontier Orbital Theory. (b) Bader charge analysis results of S-Graphene supported Pt₃₈ cluster. (c) The state of each Pt atom in Pt₃₈/S-Graphene. The value of Δe in Pt atom can be read from the color bar.

3. What is the primary function of carbon here? Can this sulfur stabilizing approach work with other supports that do not contain carbon? Can it work, for example, with TiO₂ as support?

Response: Thanks for the reviewer's valuable comments here. We believe that the primary function of carbon is to supply a thermally stable matrix to stabilize sulfur sites. According to the reviewer's comments, we tried to apply sulfur-stabilizing method to TiO₂ support. First, we synthesized the S-doped TiO₂ powder by pyrolysis of titanium tetraisopropanolate and thiourea. Typical procedures are shown in the method section of manuscript. The commercial TiO₂ powder (P-25) was selected as reference support.

Both undoped and S-doped TiO₂ were used as supports to load Pt catalysts (1.0 wt% Pt) by the impregnation method. Then we carried out the sintering test with the Pt/TiO₂ and Pt/S-TiO₂ from 300 to 700 °C to observe the change of Pt clusters. Because the diffraction of TiO₂ supports is too strong, we did not observed any Pt diffraction peaks for the fresh catalysts with the low Pt loading of 1.0 wt%. After the sintering test at 700 °C, the Pt/TiO₂ catalyst gradually appeared Pt (111) diffraction peak, while the Pt/S-TiO₂ catalyst still remained unchanged (Fig. R5). HADDF-STEM observations and EDS elemental mapping further affirmed the Pt sintering on Pt/TiO₂ and the anti-sintering on Pt/S-TiO₂ with uniform distribution of sulfur element (Fig. R6). These results demonstrated the universality of the concept of strong metal-sulfur interaction in stabilizing metal cluster catalysts at high temperatures.

Figure R5. XRD patterns of the Pt/S-TiO₂ and Pt/TiO₂ catalysts after annealing at different temperature (300°C, 400°C, 500°C, and 700°C) in 5% H₂/Ar for 120 min.

Figure R6. HADDF-STEM images and EDS elemental mappings of the Pt/S-TiO₂ and Pt/TiO₂ catalysts after annealing at 700 °C in 5% H₂/Ar for 120 min.

The above results and related discussion have been added in the revised manuscript (Page 5) and Supplementary information (Figs. S16 and S17).

4. I suggest the authors change Fig. 1 to a more demonstrative schematic illustration. The current figure is a bit confusing and does not give precise overall info of the manuscript story.

Response: Thanks for the reviewer's valuable comments here. As suggested by the reviewer, we have revised the schematic illustration to give clear description of the manuscript story (Fig. R7).

The above figure have been changed in the revised manuscript (Fig. 1).

Figure R7. Schematic illustration of the sulfur-stabilizing method for suppressing metal nanocluster sintering at high temperatures.

5. The authors also need to draw another figure representing the overall mechanism of S-C stabilizing with metal nanocluster interaction, especially demonstrating the following statement “The increased electron density of Pt nanoclusters can weaken the bond between Pt and an electron-rich group of C=C of propylene on account of electrostatic repulsion⁴¹, which would enhance the propylene desorption and accordingly improve the PDH selectivity.” This figure could also include the overall dehydrogenation process of propane using Pt-supported S-C catalyst.

Response: Many thanks for the reviewer’s valuable comments on this issue. As suggested by the reviewer, we have drawn an additional figure about the mechanism of metal-S interaction under dehydrogenation process of propane (Fig. R8).

The above figure have been added in the revised manuscript (Fig. 6b).

Figure R8. Illustration of the metal-S interaction with high selectivity under PDH reaction.

6. The authors have focused on using only single metal nanoclusters in the catalysis, but what about bimetallic nanoclusters? Such as Pt-Ir/S-C catalysts? Or even tertiary nanoclusters? Can they also work in this stabilizing method?

Response: Many thanks for the reviewer's comments on this issue. According to the reviewer's comments, we synthesized and performed the sintering tests of bimetallic nanoclusters (including Pt-Ir and Ir-Ru) and tertiary nanoclusters (Pt-Ir-Ru) catalysts (Fig. R9). Similar to the monometallic system, no any aggregation or overgrowth of bimetallic or tertiary nanoclusters were found by HAADF-STEM observations with a broad vision after the harsh thermal treatment up to 700 °C for 10 h. These results further demonstrate the generality of the sulfur-stabilizing method for synthesizing thermally stable multi-metallic nanoclusters.

The above results and related discussion have been added in the revised manuscript (Page 6) and Supplementary Information (Fig. S15).

Figure R9. Sintering tests of the Pt-Ir, Ir-Ru, Pt-Ir-Ru nanoclusters before (a) and after (b) sintering tests. Sintering test condition: 5% H₂/Ar, 700 °C, 600 min.

7. I also suggest that the authors demonstrate the applicability of their catalysis in other high-temperature reactions, such as methane or CO conversion.

Response: Many thanks for the reviewer's comments on this issue. As suggested by the reviewer, we further evaluated the stability of the Pt/S-C and Pt/S-free-C for water-gas shift (WGS) reaction (Fig. R10). The catalytic performance of the catalysts in the WGS reaction was evaluated in a fixed-bed flow reactor. The catalyst (10 mg) was pretreated by 5% H₂/Ar at 200 °C for 2 h. After that, the temperature was increased to 400 °C, and the catalyst was exposed to the WGS reaction mixture. The reactant gas consisted of 5% CO (flow rate: 30 mL min⁻¹) and water vapor at 46 °C (water vapor pressure: 10.094 kPa) balanced with Ar that yielded the P_{CO}/P_{H₂O} ratio of 1:2. All catalysts were heated to the desired reaction temperatures at a rate of 1 K min⁻¹, and the steady state compositions of the effluent gas were analyzed with an online gas chromatograph (FULI 9790II) with a TCD attached to a TDX column. The catalytic activity was calculated by the change in the CO concentrations of the inlet and outlet gases. The

WGS rate was calculated based on the total Pt content. As shown in Figure R10, Pt/S-free-C exhibited a higher initial activity ($286.7 \text{ mol}_{\text{CO}}\text{g}_{\text{Pt}}^{-1}\text{min}^{-1}$) than Pt/S-C, which could be ascribed to the high oxidation state of Pt on S-free-C. The electron-deficient Pt is beneficial to weaken CO binding and promote reaction (Nature 2017, 544 (7648), 80-83; Nat. Nanotechnol. 2019, 14 (4), 354-361). Although Pt/S-free-C exhibited a higher initial WGS activity, the rate rapidly decayed to $129.8 \text{ mol}_{\text{CO}}\text{g}_{\text{Pt}}^{-1}\text{min}^{-1}$ after 600 min, which further decayed to $70.3 \text{ mol}_{\text{CO}}\text{g}_{\text{Pt}}^{-1}\text{min}^{-1}$ after 2800 min. Encouragingly, the Pt/S-C catalyst exhibited a stable WGS rate about $129.8 \text{ mol}_{\text{CO}}\text{g}_{\text{Pt}}^{-1}\text{min}^{-1}$ after continuous operation for 3500 min. The enhanced hydrothermal stability at high temperature further demonstrated the application potentials of the sulfur-stabilizing method for industrially relevant catalysis.

The above results and related discussion have been added in the revised manuscript (Page 9) and Supplementary information (Fig. S26).

Figure R10. Stability evaluation of the Pt/S-C and Pt/S-free-C catalysts in high-temperature WGS reactions.

Reviewer #3 (Remarks to the Author):

Yin et al. reported a sulfur doped carbon matrix to stabilize the various metal nanoclusters (Pt, Ru, Rh, Os, and Ir). Among the synthesized nanoclusters, Pt was found to perform high selectivity in propane dehydrogenation. Despite the topic may be interesting to the community, several uncertainties are required to be included in order to draw a more rigorous scientific conclusion for the current experimental findings. Therefore, the current form is suggested to undergo a major revision subject to further review with the additional improvement.

1. Figure 3(c) contains the Fourier transform of the EXAFS spectra of Pt/S-C, Pt/S-free-C, PtS₂, PtO₂, and Pt foil. Both PtS₂ and PtO₂ are bulk materials, however, the second coordination feature of PtS₂ is absent in the corresponding measurement. That is consequently challenges the accuracy of Pt-S bond identity in the Pt/S-C sample.

Response: Thanks for the reviewer's valuable comments here. We re-checked the data quality of XAFS of bulk PtS₂ sample that was purchased from Sigma-Aldrich. The k space of k^3 -weighted Pt L₃-edge of PtS₂ (Fig. R11) shows the high data quality with only minor noise over 11 Å⁻¹, indicating the reliability of the PtS₂ XAFS spectrum. We noted that similar weak second coordination feature of PtS₂ was also present in previously reported work (Electrocatalysis 2019, 10 (5), 516-523.), although the author did not explain this issue there. We assume that such feature could be attributed to the low order degree over a long range, but it does not interfere with the accuracy of Pt-S bond (about 1.88 Å) in first coordination layer.

The above results and related discussion have been added in the revised manuscript (page 6) and Supplementary information (Fig. S18).

Figure R11. R-space (a) and k-space (b) of k^2 -weighted Pt L3-edge of PtS_2 .

2. Additionally, the comparable Fourier transform of the EXAFS results were not provided for the Ru, Rh, Os, and Ir cases. A consistent comparison showing the superior selectivity of Pt/S-C over other tested metal clusters was not on the same basis.

Response: Thanks for the reviewer's comments here. The reviewer may misunderstand the contents, probably owing to our writing. Actually, the other metals (Ru, Rh, Os, and Ir) were only involved in the catalyst synthesis and sintering experiments to show the generality of our sulfur-stabilizing method but not in propane dehydrogenation tests. We noted that the PDH reaction was rarely reported by other metals besides Pt. According to the reviewer's comments, we also tested the catalytic activity of other S-C supported metal cluster catalysts, but their activity and selectivity was poor (Fig. R12). For examples, Os/S-C showed a very low activity with a conversion of less than 4%, while Ru/S-C and Rh/S-C exhibited a slightly higher conversion of around 10%. Though the activity of Ir/S-C (~35% conversion) is comparable to that of Pt/S-C, the selectivity is poor (~80%). Further exploration of the catalytic applications of these metal cluster catalysts for other reactions can be carried out in future.

Figure R12. Catalytic performance of Ru/S-C (a), Rh/S-C (b), and Ir/S-C (c) under PDH reaction.

3. The selection of M_{38} cluster model to represent the synthesized M/S-C catalysts need more supportive evidences. The selection of size and shape of DFT models show any consistent evidence with any experimental observables?

Response: Many thanks for the reviewer's comment on this issue. Kumar et al. reported that 3x3x4 cuboid is the most stable configuration for Pt_{36} cluster and the most stable configuration of Pt_n cluster gradually transform from cuboid into regular octahedron from $n=36$ to $n=44$ (Physical Review B, 2008, 77, 205418.) In our previous work (Nat. Commun. 2019, 10, 1-9), we have calculated the binding energy for a series of Pt_n clusters. Our results indicate that the most stable configuration of Pt_{38} cluster is Pt_{36} cuboid capped with two Pt atoms on the 3x4 plane (about 5.26 eV/atom). And the Pt_{38} cluster binding energy in the shape of truncated octahedron (about 5.24 eV/atom) is very close to that of cuboid-shape, which means that the truncated octahedron is also energy favored.

Meanwhile, The HAADF-STEM image show that the exposed crystal faces of Pt/S-C are (111) and (200) (Fig. R13), which is in good agreement with the atomic array and exposed crystal plane of the proposed Pt_{38} structure. Overall, considering the computational and experimental results, we chose the Pt_{38} clusters with truncated octahedron shape in the current simulation work.

Figure R13. (a) Binding energy of a series of Pt_n cluster. The binding energy of cuboid-shape Pt₃₈ and truncated octahedron Pt₃₈ are marked with blue and red colors, respectively (from Nat. Commun. 2019, 10 (1), 1-9.). (b-c) HAADF-STEM image of Pt/S-C. Truncated octahedron Pt₃₈ cluster with the zone axis [01-1] perpendicular to the paper. The orange and blue lines mark the crystal face (111) and (200) of Pt₃₈ cluster, respectively.

The above results and related discussion have been added in the revised manuscript (Page 6).

4. The DFT results of Figure 4 are also confusing. Figure 4(a) should be renamed as Pt₃₈ cluster desorption since the FS geometry of Pt₃₈/S-C is substantially away from the graphene plane. The individual atom “escape energy” in Figure 4(b) is ambiguous due to the selection of the “single atom” is not well-defined, and the “escape energy” is believed to be substantially subject to the atom selection.

Response: Thanks for the reviewer’s valuable comments here. We have corrected the description of Figure 4(a) to “Energy barrier of Pt₃₈ cluster desorption on S-Graphene and Graphene” to make our manuscript more precise. Moreover, we further selected several other atoms at different positions to calculate the average escape energy of individual atom (Fig. R14). As can be seen, the escape energy of all individual atoms on S-Graphene was larger than that on Graphene, which can be ascribed to the enhanced

strength of interfacial adhesion at the Pt₃₈/S-Graphene owing to the metal-S bonding.

The above results and related discussion have been added in the revised manuscript (Page 7, Fig. 4).

Figure R14. Theoretical investigations of the energy barriers for nanoparticle desorption and atom escaping. (a) Energy barrier of Pt₃₈ cluster desorption on S-Graphene and Graphene. (b) Escape energy of individual atoms from the Pt₃₈ cluster at different positions on S-Graphene and Graphene. The higher energy barriers for both Pt₃₈ migration and individual atom escaping on the S-Graphene substrate support the experimentally observed sinter resistance of Pt nanoclusters on S-C.

5. The *C₃H₆ intermediate of Fig 5(f) show significant stability with the Pt₃₈/S-C model, and such an enhanced stability could be highly related to the observed dehydrogenation selectivity. More insightful electronic structure analysis should have been provided to show absorbed geometric effect, charged transfer effect as well as the intrinsic bonding property of metal!

Response: Thanks for the reviewer's valuable comments here. Due to the strong

interaction between $C_3H_6^*$ and Pt_{38}/S -Graphene model, that $C_3H_6^*$ can hardly change to $C_3H_5^*$, $Pt/S-C$ exhibits a higher C_3H_6 selectivity. As for the adsorbed geometric effect, we have repeatedly optimized the molecular adsorption sites before the PDH reaction paths (Fig. R15-R17), and the final system is spontaneously stabilized at the sites shown in Figure R15.

As suggested by the reviewer, we further study the charge transfer by the density of state (DOS) analyses for the two model (Fig. R18). For the $C_3H_8^*$ adsorption stage, the d-band of two systems show a small difference near the Fermi level, indicating that both are favorable for catalysis. While in the $C_3H_6^*$ adsorption stage, the d-band of Pt_{38}/S -Graphene model shift down away from Fermi level than $Pt_{38}/Graphene$ from the change of DOS, which indicated the low activity for deep cracking on Pt_{38}/S -Graphene model. Using Bader analysis, we further found that Pt_{38}/S -Graphene transferred 0.08 e to $C_3H_6^*$ intermediate but $Pt_{38}/Graphene$ only transferred 0.02 e, which may be the cause of the down-shift of d-band and the decreased reactivity for deep dehydrogenation in Pt_{38}/S -Graphene model.

The above results and related discussion have been added in the revised manuscript (Page 10) and Supplementary information (Figs. S27-S30).

Figure R15. The Optimization of molecular adsorption sites. The red dotted frame is the spontaneously final stable state.

Figure R16. The PDH reaction paths on the Pt₃₈/Graphene.

Figure R17. The PDH reaction paths on the Pt₃₈/S-Graphene.

Figure R18. Density of states (DOS) of (a) the $C_3H_8^*$ adsorption stage and (b) the $C_3H_6^*$ adsorption stage. (c) Bader charge analysis of the $C_3H_6^*$ intermediate adsorption stage on Pt₃₈/Graphene and Pt₃₈/S-Graphene.

REVIEWER COMMENTS

Reviewer #1 (Remarks to the Author):

I have carefully read the author's reply and the emended manuscript as well as the supporting information. I understand the arguments related to the novelty of this study. Nevertheless, I am not sure if it is a significant difference if the presence of S in support is decisive for stabilization of Pt nanoparticles of 3-5 nm or 1 nm as has been previously reported in several previous studies on in the present manuscript.

The practical relevance of the developed has not been justified due to the following reasons.

- i) As seen in Fig.5(c) and Figure R3(a,c), the selectivity decreases with an increase in propane concentration in reaction feeds. Nevertheless, the authors reports the values obtained with a diluted feed, where the highest propene selectivity was achieved. In addition, the selectivity was calculated on the basis of detected gas-phase products without considering coke formation. The latter is typically a major side product in propane dehydrogenation.
- ii) The fact that carbon support will burn upon oxidative catalyst regeneration is known. Why did the authors not apply their approach using an industrially relevant support?
- iii) No tests at higher reaction temperatures relevant for industrial applications have been performed.

As the authors did not provide units for terms used for calculating the rate of propene formation, it is not easy to check if this formula correct. It is also not explained if the rate was determined under integral or differential conditions.

Reviewer #3 (Remarks to the Author):

I am satisfied with the changes made by the authors and their explanations on the scientific comments/concerns risen by reviewers. It could be a even better presentation if couple sentences of the writing can adjusted to be more academic style. This manuscript is recommended for publication after minor revisions noted.

There are a few examples that can be further revised.

At page 4, the sentence - "no one has reported" could be further revised.

At page 4, the sentence - "we must clarify" could be further revised.

Reviewer #1 (Remarks to the Author):

I have carefully read the author's reply and the emended manuscript as well as the supporting information. I understand the arguments related to the novelty of this study. Nevertheless, I am not sure if it is a significant difference if the presence of S in support is decisive for stabilization of Pt nanoparticles of 3-5 nm or 1 nm as has been previously reported in several previous studies on in the present manuscript.

The practical relevance of the developed has not been justified due to the following reasons.

i) As seen in Fig.5(c) and Figure R3(a,c), the selectivity decreases with an increase in propane concentration in reaction feeds. Nevertheless, the authors reports the values obtained with a diluted feed, where the highest propene selectivity was achieved. In addition, the selectivity was calculated on the basis of detected gas-phase products without considering coke formation. The latter is typically a major side product in propane dehydrogenation.

Response: Thanks for the reviewer's valuable comment here. According to the reviewer's comments, we have deleted the high selectivity values in the Abstract section and further supplemented the description of selectivity changes under different propane concentration feeding (Figure R1), which make the discussion more objective. Though the selectivity decreased with an increased propane concentration in reaction feeding, the S-contained catalyst was still superior to S-free catalyst, which further suggested the positive role of strong metal-sulfur interaction in promoting the selectivity of propane dehydrogenation.

As for the calculation of selectivity, we took the experiment of Pt/S-C under 33% C₃H₈ feeding as example (Table R1). The reaction product gas (CH₄, C₂H₄, C₂H₆, C₃H₆, and C₃H₈) were continuously detected by online flame ionization detector (FID) and thermal conductivity detector (TCD) gas chromatograph equipped downstream. Under sampling at intervals of six minutes, we could obtain the corresponding peak area of

the reaction product (column 2-6 of Table R1). Using the external standard method, we could quantify the molar content of each product by peak area and obtain the relative molar ratio (column 7-11 of Table R1), and further calculate the carbon balance (Eqs. 1, column 12 of Table R1). Carbon balance typically ranged between 95% and 105% for all the reactions, which allows for ignoring the loss of carbon deposition to some extent.

Figure R1. The change of PDH selectivity under different propane concentration in reaction feeding.

The carbon balance were determined by following equation:

$$\text{Carbon balance} = \frac{\frac{1}{3}[\text{CH}_4]_{\text{outlet}} + \frac{2}{3}[\text{C}_2\text{H}_4]_{\text{outlet}} + \frac{2}{3}[\text{C}_2\text{H}_6]_{\text{outlet}} + [\text{C}_3\text{H}_6]_{\text{outlet}} + [\text{C}_3\text{H}_8]_{\text{outlet}}}{[\text{C}_3\text{H}_8]_{\text{inlet}}}$$

$$= \frac{1}{3}C_{\text{CH}_4} + \frac{2}{3}C_{\text{C}_2\text{H}_4} + \frac{2}{3}C_{\text{C}_2\text{H}_6} + C_{\text{C}_3\text{H}_6} + \frac{1}{3}C_{\text{C}_3\text{H}_8}$$

where, [CH₄], [C₂H₄], [C₂H₆], [C₃H₆], and [C₃H₈] are the molar content of each product. C_{CH₄}, C_{C₂H₄}, C_{C₂H₆}, C_{C₃H₆}, and C_{C₃H₈} are the molar ratio in column 7-11 of Table R1.

The above results and discussion catalyst have been added in the revised manuscript (page 9) and Supplementary Information (Fig. S26).

Table R1. The raw catalytic data of Pt/S-C under 33% C₃H₈ feeding.

Time(min)	Peak area					Molar ratio					Carbon balance	Conv.%	Sel.%
	CH ₄	C ₂ H ₄	C ₂ H ₆	C ₃ H ₆	C ₃ H ₈	CH ₄	C ₂ H ₄	C ₂ H ₆	C ₃ H ₆	C ₃ H ₈			
6	599457	128840	1344788	24765685	159680843	0.011319392	0.001313	0.01356	0.171522	0.812886	0.997958734	18.55	92.85
12	677621	125928	1222025	25044400	161206350	0.012795343	0.001283	0.012322	0.173452	0.820652	1.007305445	18.53	93.12
18	723699	120604	1204429	24800908	160639417	0.013665422	0.001229	0.012144	0.171766	0.817765	1.002867216	18.46	93.00
24	553062	118157	1200481	25053412	161336251	0.010443328	0.001204	0.012105	0.173515	0.821313	1.007057351	18.44	93.57
30	669512	115231	1204434	25098786	161781190	0.012642223	0.001174	0.012145	0.173829	0.823578	1.010368915	18.49	93.25
36	664727	112522	1196214	25062395	162343110	0.012551869	0.001146	0.012062	0.173577	0.826438	1.012874702	18.41	93.29
42	730227	108563	1109176	24956243	160983843	0.013788689	0.001106	0.011184	0.172842	0.819519	1.005022209	18.46	93.38
48	654713	107517	1013829	25081191	161780156	0.012362777	0.001095	0.010223	0.173707	0.823573	1.008829245	18.36	93.95
54	587688	105539	1309039	25083498	162246882	0.011097161	0.001075	0.013199	0.173723	0.825949	1.012754818	18.45	93.16
60	666837	103522	1079142	25279290	162758931	0.012591712	0.001055	0.010881	0.175079	0.828555	1.015667212	18.42	93.76
66	718001	101230	101230	25102225	162293259	0.013557828	0.001031	0.001021	0.173853	0.826185	1.005865809	17.86	96.98
72	735135	100994	1112269	25430307	163198279	0.013881365	0.001029	0.011215	0.176125	0.830792	1.019578751	18.52	93.50
78	665267	99506	1181920	25347194	162526618	0.012562066	0.001014	0.011918	0.175549	0.827373	1.015602055	18.53	93.45
84	656181	97643	1165762	25252007	162537535	0.012390497	0.000995	0.011755	0.17489	0.827428	1.01482171	18.47	93.51
90	539395	96244	1160537	25275015	162536113	0.010185257	0.000981	0.011702	0.175049	0.827421	1.014201911	18.42	93.87
96	655227	93147	1265073	25238443	162317950	0.012372483	0.000949	0.012756	0.174796	0.82631	1.014234656	18.53	93.20
102	570578	92433	1270870	25197014	161952559	0.010774078	0.000942	0.012814	0.174509	0.82445	1.011593939	18.50	93.41
108	696458	83578	1253705	25201855	161360174	0.013151037	0.000852	0.012641	0.174543	0.821435	1.009222438	18.61	93.14
114	568854	45650	1057073	25082769	161474450	0.010741524	0.000465	0.010659	0.173718	0.822016	1.006620665	18.34	94.27
120	635198	69185	1046899	25092998	161570715	0.01199428	0.000705	0.010556	0.173789	0.822506	1.007685528	18.38	94.03
126	532826	88190	1265761	25351680	162339235	0.010061216	0.000899	0.012763	0.17558	0.826419	1.014335794	18.53	93.59
132	647084	86682	1135129	25245539	161643735	0.012218721	0.000883	0.011446	0.174845	0.822878	1.009892603	18.52	93.68
138	644156	84907	1029502	25291785	162044310	0.012163432	0.000865	0.010381	0.175166	0.824917	1.011518978	18.45	94.06
144	522622	83910	946315	25106985	161457219	0.009868537	0.000855	0.009542	0.173886	0.821929	1.005932773	18.29	94.65
150	560521	79832	1240387	25174742	160857761	0.010584174	0.000813	0.012507	0.174355	0.818877	1.005516145	18.56	93.58
156	523434	82781	1014061	25245055	161864828	0.00988387	0.000843	0.010225	0.174842	0.824004	1.009412338	18.37	94.45
162	520935	80425	1095102	25112393	161167694	0.009836682	0.000819	0.011042	0.173923	0.820455	1.005452561	18.40	94.16
168	519669	80760	1008738	25113178	161257860	0.009812776	0.000823	0.010171	0.173929	0.820914	1.005336625	18.34	94.46
174	518202	79466	937540	25075560	160689462	0.009785075	0.00081	0.009453	0.173668	0.81802	1.001690895	18.34	94.71
180	521962	79454	941700	25183260	161080846	0.009856074	0.00081	0.009495	0.174414	0.820013	1.00448025	18.36	94.70
186	687810	79132	1219740	25279599	161765183	0.012987739	0.000806	0.012299	0.175081	0.823496	1.011512832	18.59	93.31
192	627234	78484	1022632	25293533	162155620	0.011843898	0.0008	0.010311	0.175178	0.825484	1.011903372	18.42	94.15
198	548092	77713	1213566	25255646	161404095	0.01034948	0.000792	0.012237	0.174915	0.821658	1.008587497	18.53	93.73

204	544202	76963	1014431	25252501	161474223	0.010276026	0.000784	0.010229	0.174893	0.822015	1.007568209	18.42	94.41
210	513685	75348	924026	25073495	160835358	0.009699782	0.000768	0.009317	0.173654	0.818763	1.002273532	18.31	94.78
216	536200	75409	924792	25109800	160972861	0.010124927	0.000768	0.009325	0.173905	0.819463	1.003370763	18.33	94.72
222	515437	75480	1020199	25265023	161398498	0.009732864	0.000769	0.010287	0.17498	0.82163	1.007118611	18.42	94.48
228	518271	75235	996078	25386804	162628048	0.009786378	0.000767	0.010044	0.175824	0.827889	1.014076787	18.36	94.58
234	514999	74056	1079921	25232426	161605181	0.009724594	0.000755	0.010889	0.174754	0.822682	1.008330149	18.41	94.28
240	546118	73087	1074173	25310600	161557805	0.010312206	0.000745	0.010831	0.175296	0.822441	1.008779533	18.47	94.23
246	539867	73106	1193673	25260554	162075871	0.01019417	0.000745	0.012036	0.174949	0.825078	1.011826674	18.46	93.84
252	510061	71597	1004858	25189321	161212689	0.009631351	0.000729	0.010132	0.174456	0.820684	1.005486716	18.38	94.55
258	534292	71896	991986	25315854	161986757	0.010088898	0.000733	0.010002	0.175332	0.824624	1.010370936	18.38	94.55
264	511667	71637	988677	25252025	161779254	0.009661676	0.00073	0.009969	0.17489	0.823568	1.00870779	18.35	94.61
270	511058	71133	915868	25338886	161912578	0.009650177	0.000725	0.009235	0.175492	0.824247	1.009496362	18.35	94.88
276	512077	70654	993646	25373842	162155206	0.009669418	0.00072	0.010019	0.175734	0.825482	1.011494337	18.39	94.62
282	513206	70087	915754	25347792	162003802	0.009690737	0.000714	0.009234	0.175553	0.824711	1.010028029	18.35	94.88
288	509721	64075	1007897	25334573	161878187	0.009624931	0.000653	0.010163	0.175462	0.824072	1.009848068	18.40	94.59
294	537068	68729	994176	25291994	161701215	0.010141317	0.0007	0.010024	0.175167	0.823171	1.008762657	18.40	94.54
300	507225	68320	988367	25147142	161206439	0.009577799	0.000696	0.009966	0.174164	0.820652	1.005013322	18.34	94.62
306	508231	67899	908608	25348582	161759182	0.009596795	0.000692	0.009162	0.175559	0.823466	1.008694945	18.36	94.93
312	543267	67319	996192	25266871	161630907	0.010258371	0.000686	0.010045	0.174993	0.822813	1.008273306	18.39	94.51
318	506429	66824	903610	25221986	161288689	0.009562769	0.000681	0.009111	0.174682	0.821071	1.005371316	18.33	94.93
324	508199	66694	906086	25318395	161445429	0.009596191	0.00068	0.009136	0.17535	0.821869	1.006863571	18.37	94.93
330	508108	66660	994146	25358166	162216687	0.009594473	0.000679	0.010024	0.175625	0.825795	1.011650487	18.37	94.64
336	503960	66112	896414	25293621	161179382	0.009516147	0.000674	0.009039	0.175178	0.820514	1.005242935	18.38	94.98
342	508742	65964	908018	25419064	162273355	0.009606444	0.000672	0.009156	0.176047	0.826083	1.01178683	18.35	94.95
348	505160	65322	896294	25279980	161223744	0.009538806	0.000666	0.009037	0.175084	0.82074	1.00537566	18.36	94.97
354	523679	64904	971872	25256231	161410497	0.009888496	0.000661	0.0098	0.174919	0.821691	1.006777431	18.38	94.66
360	507251	64811	971060	25450610	161592581	0.00957829	0.00066	0.009791	0.176265	0.822618	1.008942195	18.47	94.75
366	536138	64426	899913	25372196	161547860	0.010123756	0.000656	0.009074	0.175722	0.82239	1.007875395	18.40	94.89
372	504684	63647	985600	25450347	161916842	0.009529818	0.000648	0.009938	0.176264	0.824268	1.010664025	18.44	94.71
378	522128	63279	896681	25425956	161962634	0.009859209	0.000645	0.009041	0.176095	0.824502	1.010242689	18.39	94.96
384	503923	63091	950926	25433623	161940785	0.009515448	0.000643	0.009588	0.176148	0.82439	1.010430852	18.41	94.83
390	507747	63282	969462	25549912	162130020	0.009587656	0.000645	0.009775	0.176953	0.825354	1.012348049	18.47	94.78
396	503089	62062	975778	25379634	162126715	0.0094997	0.000632	0.009839	0.175774	0.825337	1.011156719	18.38	94.74
402	504899	62516	895366	25494034	162590250	0.009533878	0.000637	0.009028	0.176566	0.827697	1.01378794	18.36	95.03
408	650604	61933	1149435	25430832	162447139	0.012285188	0.000631	0.01159	0.176129	0.826968	1.015216497	18.54	93.75
414	500543	61706	887036	25355380	162254915	0.009451625	0.000629	0.008944	0.175606	0.825989	1.011032539	18.30	95.05

420	505095	61838	895036	25549591	163261995	0.009537579	0.00063	0.009025	0.176951	0.831116	1.017586827	18.32	95.04
426	620997	61655	1056901	25585538	163328690	0.011726127	0.000628	0.010657	0.1772	0.831456	1.019973497	18.48	94.17
432	503862	61022	1159412	25470783	163476971	0.009514297	0.000622	0.011691	0.176405	0.832211	1.01988162	18.40	94.14
438	522093	60836	969617	25490712	163295689	0.009858548	0.00062	0.009777	0.176543	0.831288	1.01794608	18.34	94.73
444	650476	60074	1145262	25347243	162523729	0.012282771	0.000612	0.011548	0.17555	0.827358	1.014986404	18.49	93.75
450	501778	58858	956113	25393428	162938365	0.009474945	0.0006	0.009641	0.175869	0.829469	1.015223528	18.30	94.82
456	508159	58815	869324	25223065	161992000	0.009595436	0.000599	0.008766	0.17469	0.824651	1.008687881	18.25	95.07
462	495312	58696	955842	24588122	161765416	0.009352849	0.000598	0.009638	0.170292	0.823498	1.003631846	17.95	94.68
468	532827	59886	987924	25364556	163199329	0.010061235	0.00061	0.009961	0.17567	0.830797	1.016764141	18.29	94.62
474	516092	60219	966128	25497530	163089540	0.009745233	0.000614	0.009742	0.17659	0.830238	1.016879097	18.35	94.76
480	501929	59869	984915	25517429	163544203	0.009477796	0.00061	0.009931	0.176728	0.832553	1.01936588	18.33	94.75
486	534074	59465	876867	25444068	162960910	0.010084782	0.000606	0.008842	0.17622	0.829583	1.015366974	18.30	95.01
492	501249	59300	884518	25428658	162829129	0.009464956	0.000604	0.008919	0.176113	0.828913	1.014434656	18.29	95.07
498	601888	59072	974255	25348385	162905703	0.011365296	0.000602	0.009824	0.175558	0.829302	1.015491286	18.33	94.46
504	499014	58752	971526	25507867	162903057	0.009422753	0.000599	0.009796	0.176662	0.829289	1.015921002	18.37	94.80
510	499370	58682	976946	25376006	162791854	0.009429475	0.000598	0.009851	0.175749	0.828723	1.014479478	18.31	94.76
516	528610	58569	978153	25407381	162853035	0.009981607	0.000597	0.009863	0.175966	0.829034	1.015197704	18.34	94.67
522	500832	58608	973066	25485839	162763103	0.009457082	0.000597	0.009812	0.176509	0.828576	1.015076587	18.37	94.79
528	498802	58088	968632	25408538	163249417	0.00941875	0.000592	0.009767	0.175974	0.831052	1.016971237	18.28	94.79
534	513631	57962	964210	25388719	162180421	0.009698762	0.000591	0.009722	0.175837	0.82561	1.011454176	18.37	94.76
540	492026	57331	866430	25196759	161613081	0.0092908	0.000584	0.008736	0.174507	0.822722	1.006446964	18.25	95.13
546	520129	57261	964266	25182714	161954288	0.009821462	0.000583	0.009723	0.17441	0.824459	1.008912406	18.28	94.71
552	517774	56975	962448	25200299	161664287	0.009776993	0.00058	0.009705	0.174532	0.822983	1.007529194	18.32	94.72
558	498166	57269	961989	25402758	162668716	0.00940674	0.000583	0.0097	0.175934	0.828096	1.013921358	18.33	94.82
564	497618	57020	877428	25367793	162488417	0.009396393	0.000581	0.008847	0.175692	0.827178	1.012193517	18.28	95.11
570	645027	56501	957109	25296115	161827049	0.012179879	0.000576	0.009651	0.175195	0.823811	1.009775601	18.42	94.39
576	493602	56373	870115	25293299	162206084	0.00932056	0.000574	0.008774	0.175176	0.825741	1.010162275	18.26	95.13
582	645302	56220	956546	25317246	162146694	0.012185072	0.000573	0.009645	0.175342	0.825439	1.01154524	18.40	94.40
588	495787	56221	872083	25420512	162639137	0.009361818	0.000573	0.008793	0.176057	0.827945	1.013273554	18.29	95.14
594	508161	56115	872289	25379513	162350596	0.009595473	0.000572	0.008795	0.175773	0.826477	1.011598495	18.30	95.10
600	497192	56186	872726	25456134	162631450	0.009388349	0.000572	0.0088	0.176304	0.827906	1.013493931	18.31	95.14
606	654674	56352	1147298	25604763	163438472	0.012362041	0.000574	0.011568	0.177333	0.832015	1.021441285	18.55	93.80
612	497842	55743	871394	25445827	163125944	0.009400622	0.000568	0.008786	0.176232	0.830424	1.01593207	18.26	95.14
618	606947	55146	1122342	25399320	162489542	0.011460824	0.000562	0.011317	0.17591	0.827184	1.014716132	18.48	93.97
624	608538	55171	952907	25417423	162649064	0.011490867	0.000562	0.009608	0.176036	0.827996	1.014536096	18.39	94.54
630	613312	55101	1132880	25382690	162627746	0.011581013	0.000561	0.011423	0.175795	0.827887	1.015413999	18.47	93.92

636	612208	54958	876196	25591206	163861642	0.011560166	0.00056	0.008835	0.177239	0.834169	1.021423471	18.33	94.83
642	612357	54752	1020815	25536295	163706599	0.01156298	0.000558	0.010293	0.176859	0.83338	1.021215861	18.39	94.33
648	616149	55061	956005	25703883	164489183	0.011634583	0.000561	0.00964	0.17802	0.837363	1.025954836	18.38	94.57
654	502245	54619	877974	25629296	164131140	0.009483763	0.000556	0.008853	0.177503	0.835541	1.022383545	18.28	95.15
660	615219	54422	945714	25556017	163438099	0.011617022	0.000554	0.009536	0.176996	0.832013	1.019501427	18.39	94.58
666	499105	53840	869632	25302488	162335919	0.009424471	0.000549	0.008769	0.17524	0.826402	1.010900909	18.25	95.13
672	605162	53046	864357	25213066	162083824	0.011427118	0.00054	0.008715	0.17462	0.825118	1.009618684	18.27	94.82
678	601378	52960	861745	25186935	161738912	0.011355666	0.00054	0.008689	0.174439	0.823363	1.007640326	18.29	94.84
684	603781	53409	999276	25296637	162920963	0.011401041	0.000544	0.010076	0.175199	0.82938	1.015350801	18.32	94.38
690	609017	53495	1014627	25426250	163328068	0.011499911	0.000545	0.010231	0.176097	0.831453	1.018456283	18.36	94.34
696	498159	53579	1136448	25497138	163827350	0.009406608	0.000546	0.011459	0.176588	0.833994	1.02160941	18.36	94.26
702	607740	53355	1120537	25327711	163037927	0.011475798	0.000544	0.011299	0.175414	0.829976	1.016992727	18.39	93.97
708	490692	52921	931858	25357524	163379245	0.009265611	0.000539	0.009396	0.175621	0.831713	1.01694883	18.21	94.95
714	607481	53036	956746	25474845	163810513	0.011470907	0.00054	0.009647	0.176433	0.833909	1.02085097	18.31	94.55
720	640213	52026	943586	25106216	162557367	0.012088977	0.00053	0.009514	0.17388	0.827529	1.012028139	18.23	94.43
726	492533	52265	865511	25280399	162714714	0.009300374	0.000533	0.008727	0.175087	0.82833	1.012597285	18.20	95.16
732	494615	52554	949178	25378531	163370398	0.009339688	0.000535	0.009571	0.175766	0.831668	1.017186523	18.24	94.89
738	608633	52766	1014891	25297877	163143451	0.01149266	0.000538	0.010233	0.175208	0.830513	1.016621833	18.31	94.32
744	490206	51904	855201	25204341	162968346	0.009256434	0.000529	0.008623	0.17456	0.829621	1.013276146	18.12	95.19
750	493214	52224	1002930	25324935	163238436	0.009313233	0.000532	0.010113	0.175395	0.830996	1.016490316	18.25	94.70
756	490725	52307	935441	25295393	163138666	0.009266234	0.000533	0.009432	0.17519	0.830488	1.015313732	18.20	94.93
762	605257	52029	1112795	25228974	162627518	0.011428912	0.00053	0.011221	0.17473	0.827886	1.014143719	18.37	93.98
768	603803	51702	1106585	25202605	162711219	0.011401457	0.000527	0.011158	0.174548	0.828312	1.014334603	18.34	94.00
774	490348	51805	1115280	25108157	162608604	0.009259115	0.000528	0.011246	0.173894	0.82779	1.012509679	18.24	94.28
780	635739	51883	943202	25195954	162098451	0.012004496	0.000529	0.00951	0.174502	0.825193	1.010282051	18.32	94.46
786	626529	51438	1227017	25206745	162586733	0.011830586	0.000524	0.012372	0.174577	0.827679	1.014670856	18.43	93.54
792	487727	50995	925886	24998883	161861947	0.009209624	0.00052	0.009336	0.173137	0.823989	1.006669695	18.15	94.92
798	488538	50935	942948	25116284	162856883	0.009224937	0.000519	0.009508	0.17395	0.829054	1.012665896	18.13	94.88
804	491724	51224	996262	25263438	163179303	0.009285098	0.000522	0.010045	0.174969	0.830695	1.015702989	18.21	94.72
810	600419	50664	1109390	25093436	162881703	0.011337558	0.000516	0.011186	0.173792	0.82918	1.014437003	18.26	93.98
816	489788	50788	1114544	25078425	162288052	0.009248541	0.000517	0.011238	0.173688	0.826158	1.010656704	18.26	94.28
822	485072	50478	852128	25059201	162187061	0.00915949	0.000514	0.008592	0.173555	0.825644	1.008231626	18.11	95.20
828	599299	50242	997952	24920088	161635282	0.011316409	0.000512	0.010063	0.172591	0.822835	1.006139864	18.22	94.33
834	485341	50385	995792	25031641	162222738	0.009164569	0.000513	0.010041	0.173364	0.825826	1.009179492	18.17	94.69
840	484769	50221	922408	25049446	161977061	0.009153768	0.000512	0.009301	0.173487	0.824575	1.00755911	18.16	94.95
846	597011	49744	991979	24850086	160951021	0.011273205	0.000507	0.010002	0.172106	0.819352	1.002114328	18.24	94.34

852	486258	49847	933616	24909734	161389892	0.009181885	0.000508	0.009414	0.17252	0.821586	1.003683751	18.14	94.88
858	479721	49491	915001	24866458	161029486	0.009058448	0.000504	0.009226	0.17222	0.819751	1.001382307	18.14	94.96
864	473708	49722	908612	24908254	160950259	0.008944906	0.000507	0.009162	0.172509	0.819348	1.001190023	18.16	95.01
870	586843	48365	887880	24432000	157697173	0.011081205	0.000493	0.008953	0.169211	0.802787	0.981889021	18.24	94.65
876	668695	48458	1068974	24694244	159383973	0.012626796	0.000494	0.010779	0.171027	0.811374	0.994008075	18.37	93.84
882	585623	48104	970887	24545234	158648654	0.011058168	0.00049	0.00979	0.169995	0.807631	0.988059996	18.26	94.39
888	583918	47795	971111	24438086	158387195	0.011025973	0.000487	0.009792	0.169253	0.8063	0.985975694	18.22	94.37
894	593140	48469	986527	24723551	160387532	0.01120011	0.000494	0.009947	0.17123	0.816483	0.998300448	18.21	94.35
900	596852	48911	990930	24966213	160567497	0.011270203	0.000498	0.009992	0.172911	0.817399	1.000952624	18.34	94.38
906	677995	49011	1087435	24939120	160627504	0.012802405	0.000499	0.010965	0.172723	0.817705	1.002218991	18.41	93.80
912	592898	48372	841771	24916270	161600802	0.01119554	0.000493	0.008488	0.172565	0.82266	1.004846064	18.13	94.89
918	603099	49422	1110393	25018984	161892432	0.011388163	0.000504	0.011196	0.173276	0.824144	1.008900315	18.31	93.96
924	601617	48819	992743	24696404	162652995	0.011360179	0.000497	0.01001	0.171042	0.828016	1.009741747	18.00	94.30
930	596076	48992	932084	24862538	161414763	0.01125555	0.000499	0.009398	0.172193	0.821713	1.004151857	18.17	94.56
936	617281	49098	1097841	25029492	161641497	0.011655958	0.0005	0.01107	0.173349	0.822867	1.00769832	18.34	93.97
942	714413	48468	919609	24821035	161949315	0.013490077	0.000494	0.009273	0.171905	0.824434	1.007236505	18.15	94.25
948	596928	48596	985250	24877969	161086300	0.011271638	0.000495	0.009934	0.1723	0.82004	1.002943086	18.24	94.38
954	601163	49319	998144	25041802	162227694	0.011351606	0.000502	0.010064	0.173434	0.825851	1.010005301	18.23	94.35
960	598280	48831	981277	25068562	161896018	0.011297167	0.000498	0.009894	0.17362	0.824162	1.008368681	18.27	94.43
966	599435	48985	848565	25037764	162446943	0.011318977	0.000499	0.008556	0.173406	0.826967	1.010085017	18.13	94.87
972	593303	48196	1093552	24847228	161152695	0.011203188	0.000491	0.011026	0.172087	0.820378	1.003763637	18.27	94.01
978	480832	48262	842752	24814760	161405959	0.009079427	0.000492	0.008498	0.171862	0.821668	1.002458613	18.03	95.20
984	600067	49017	847812	24985899	162345972	0.011330911	0.000499	0.008549	0.173047	0.826453	1.009210941	18.11	94.86
990	600894	48652	1108966	24972307	162121266	0.011346527	0.000496	0.011182	0.172953	0.825309	1.009713549	18.26	93.96
996	675158	48208	1090371	24857887	162100978	0.012748835	0.000491	0.010994	0.17216	0.825206	1.009153848	18.23	93.79
1002	601418	48560	992916	24954256	162412899	0.011356421	0.000495	0.010012	0.172828	0.826794	1.01030349	18.16	94.35
1008	602403	48402	1109380	25004135	162173998	0.011375021	0.000493	0.011186	0.173173	0.825578	1.010212902	18.28	93.97
1014	596778	48371	932208	24887872	161786014	0.011268805	0.000493	0.0094	0.172368	0.823602	1.006218262	18.15	94.56
1020	596042	47983	1101520	24830109	161850888	0.011254908	0.000489	0.011107	0.171968	0.823933	1.00726802	18.20	93.97
1026	484284	48102	838545	24775457	161479669	0.00914461	0.00049	0.008455	0.17159	0.822043	1.002554081	18.01	95.20
1032	592265	47497	1090309	24756016	161433953	0.011183588	0.000484	0.010994	0.171455	0.82181	1.004530969	18.19	94.01
1038	711386	47299	955416	24517065	160659572	0.013432919	0.000482	0.009634	0.1698	0.817868	0.99877716	18.11	94.07
1044	669818	47093	1075766	24454903	159397696	0.012648001	0.00048	0.010847	0.169369	0.811444	0.992463329	18.24	93.76
1050	592967	47698	837926	24590590	160020314	0.011196843	0.000486	0.008449	0.170309	0.814614	0.994515009	18.09	94.85
1056	593417	47493	908007	24525609	159945304	0.011205341	0.000484	0.009156	0.169859	0.814232	0.994150919	18.10	94.59
1062	594351	47001	979302	24639677	161009686	0.011222977	0.000479	0.009874	0.170649	0.81965	1.000836341	18.10	94.36

1068	488343	927732	927732	24598168	160793462	0.009221255	0.009452	0.009354	0.170362	0.81855	1.004366861	18.50	91.82
1074	590374	46796	1078673	24547339	161293941	0.01114788	0.000477	0.010876	0.17001	0.821097	1.002279025	18.08	94.01
1080	594845	47145	917524	24695297	162010180	0.011232305	0.00048	0.009252	0.171034	0.824744	1.005907678	18.01	94.58
1086	601868	47696	849994	24851185	162026614	0.011364919	0.000486	0.008571	0.172114	0.824827	1.006669048	18.06	94.83
1092	680547	47361	986366	24728381	161556950	0.012850594	0.000483	0.009946	0.171263	0.822436	1.004823182	18.15	94.10
1098	595480	46796	1090782	24619846	161436610	0.011244296	0.000477	0.010999	0.170512	0.821824	1.003619878	18.11	93.97
1104	594663	46959	977300	24604308	160915224	0.011228868	0.000478	0.009854	0.170404	0.81917	1.000098845	18.09	94.36
1110	708053	46830	1058187	24434861	159859465	0.013369983	0.000477	0.01067	0.169231	0.813795	0.994794744	18.19	93.71
1116	477941	46718	836542	24490578	160504987	0.009024837	0.000476	0.008435	0.169617	0.817081	0.995557105	17.93	95.18
1122	489866	46336	931022	24448976	160055475	0.009250014	0.000472	0.009388	0.169328	0.814793	0.993681146	18.00	94.80
1128	490137	46680	1090461	24444445	160168314	0.009255131	0.000476	0.010995	0.169297	0.815367	0.995289246	18.08	94.24
1134	483668	46197	827872	24436407	161297167	0.009132978	0.000471	0.008348	0.169241	0.821114	0.999189156	17.82	95.19
1140	486751	46891	944008	24645182	161447241	0.009191194	0.000478	0.009519	0.170687	0.821878	1.002195818	17.99	94.80
1146	484665	47005	924955	24734805	162417814	0.009151805	0.000479	0.009326	0.171308	0.826819	1.007618387	17.94	94.89
1152	598215	46665	1091208	24619176	162245502	0.01129594	0.000475	0.011003	0.170507	0.825942	1.007752053	18.04	93.96
1158	594389	46705	1107604	24623964	161601766	0.011223695	0.000476	0.011168	0.17054	0.822664	1.004593695	18.11	93.91
1164	482112	46732	1100062	24601083	161309064	0.009103597	0.000476	0.011092	0.170382	0.821174	1.002195529	18.06	94.26
1170	485676	46821	932931	24611001	161733397	0.009170895	0.000477	0.009407	0.170451	0.823335	1.003334934	17.94	94.84
1176	594630	46166	1100862	24566693	161733397	0.011228245	0.00047	0.0111	0.170144	0.823335	1.00482015	18.06	93.92
1182	599582	46850	1098420	24684268	162080472	0.011321753	0.000477	0.011076	0.170958	0.825101	1.007420507	18.10	93.94
1188	482760	46519	836813	24542323	161116006	0.009115833	0.000474	0.008438	0.169975	0.820192	0.999056482	17.90	95.18
1194	485108	46474	941357	24568716	161507767	0.00916017	0.000474	0.009492	0.170158	0.822186	1.001943666	17.94	94.80
1200	622681	46561	966779	24573168	161581701	0.011757925	0.000474	0.009748	0.170189	0.822562	1.0033779	18.02	94.31
1206	479072	46187	833287	24438448	161441614	0.009046193	0.000471	0.008402	0.169255	0.821849	0.999945956	17.81	95.18
1212	595250	46407	837788	24527025	160676142	0.011239953	0.000473	0.008448	0.169869	0.817952	0.997418015	17.99	94.83
1218	599028	46581	951576	24742896	162408585	0.011311292	0.000475	0.009595	0.171364	0.826772	1.008514381	18.02	94.47
1224	484100	46826	840975	24568539	161193891	0.009141136	0.000477	0.00848	0.170156	0.820588	0.999672649	17.91	95.16
1230	631254	46907	974390	24750639	162171628	0.011919807	0.000478	0.009825	0.171418	0.825565	1.00771656	18.08	94.29
1236	488351	47090	657193	24715353	162017801	0.009221406	0.00048	0.006627	0.171173	0.824782	1.003688936	17.82	95.83

ii) The fact that carbon support will burn upon oxidative catalyst regeneration is known. Why did the authors not apply their approach using an industrially relevant support?

Response: Many thanks for the reviewer's comments on this issue. According to the reviewer's comments, we tried to apply sulfur-doped TiO₂ support to regeneration for PDH reaction under oxidation conditions. The sulfur-doped TiO₂ support Pt nanoclusters also exhibited outstanding thermal stability up to 700 °C owing to the strong Pt-S interactions (Fig. S16). After 5 successive regeneration cycles, the propane conversion and propylene selectivity remained little change (Figure R2), indicating the good regenerability and the feasibility of anti-sintering under oxidation conditions of Pt-S interactions on industrially relevant metal oxide supports.

Figure R2. Regeneration properties of Pt/S-TiO₂ during the continuous cycles. The regeneration process was carried out under flowing air stream (25 mL min⁻¹) at 500 °C for 30 min. Catalytic conditions: 0.1 wt% Pt/S-TiO₂, atmospheric pressure, 550 °C, C₃H₈/H₂= 1/1, with balance Ar for total flow rate of 20 mL min⁻¹, WHSV = 2 h⁻¹ over 120 mg of sample.

The above results and discussion have been added in the revised manuscript (page 9) and Supplementary Information (Fig. S28).

iii) No tests at higher reaction temperatures relevant for industrial applications have been performed.

Response: Thanks for the reviewer's comments on this issue. According to the reviewer's comments, we further tried a higher temperature PDH reaction at 600 °C. When the reaction temperature increased to 600 °C, the C₃H₈ conversion of Pt/S-C gradually decreased from 55% to 40% after 1200 min with a 0.030 h⁻¹ deactivation rate and the selectivity was around 92%, indicating a slightly worse catalytic performance than that under 550 °C. Under the same high-temperature condition, the Pt/S-free-C showed rapid deactivation at the beginning of the reaction, which further demonstrated the pivotal role of the Pt-S interactions in catalyzing PDH reaction.

Figure R3. The PDH reaction under a higher temperature of 600 °C. Catalytic conditions: atmospheric pressure, 600 °C, C₃H₈/H₂= 1/1, with balance Ar for total flow rate of 15 mL min⁻¹, WHSV = 2 h⁻¹ over 80 mg of sample.

The above results and discussion catalyst have been added in the revised manuscript (page 9) and Supplementary Information (Fig. S27).

As the authors did not provide units for terms used for calculating the rate of propene formation, it is not easy to check if this formula correct. It is also not explained if the rate was determined under integral or differential conditions.

Response: Thanks for the reviewer's comments on this issue. The propene formation rate was determined under differential conditions. Each time node could correspond to the yield of propene at a current time. Then the propene formation was calculated by

the formula that was defined as the moles of C₃H₆ formation per g_{pt} per hour. These details have been described in the Experimental Section in the revised manuscript.

Reviewer #3 (Remarks to the Author):

I am satisfied with the changes made by the authors and their explanations on the scientific comments/concerns risen by reviewers. It could be a even better presentation if couple sentences of the writing can adjusted to be more academic style. This manuscript is recommended for publication after minor revisions noted.

There are a few examples that can be further revised.

At page 4, the sentence - "no one has reported" could be further revised.

At page 4, the sentence - "we must clarify" could be further revised.

Response: Many thanks for the reviewer's comments on this issue. According to the reviewer's comments, we have adjusted the expression of the above sentences. We also checked the whole manuscript again and revised accordingly.

REVIEWER COMMENTS

Reviewer #1 (Remarks to the Author):

The authors have provided some additional data and revised their manuscript. The reported results show that the presence of S in the support is important for higher propene selectivity and on-stream stability in comparison with a S-free reference material. For industrially relevant feeds, the selectivity values are rather “standard” when considering relative low degree of propane conversion. For a feed with 33vol% C₃H₈, the selectivity is about 93% at only 18% propane conversion (Table R1). The selectivity becomes even lower, when performing PDH with a feed containing 50 % propane. Although, the degree of propane conversion is not given, I assume it is not higher than 18%. These values are not surprising when even comparing with the state-of-the-art catalysts based on metal oxides (DOI: 10.1039/d0cs01140a). In addition, the space time yield of propene formation calculated on the basis of Figure 5(a) is also not significantly higher than the corresponding values reported for metal oxide catalysts.

In terms of the rates of propene formation and deactivation, this catalyst does not show any unexpected results with industrially relevant feeds when comparing with previously tested Pt-based catalysts (Fig.8 and Table 2 in DOI: 10.1039/d0cs00814a).

Figure R2. It is not written which feed was used. In the case of feed with 10vol% C₃H₈, 0.1 wt% Pt/S-TiO₂ deactivates significantly quicker in comparison with Pt/S-C

Reviewer #1 (Remarks to the Author):

Comments: The authors have provided some additional data and revised their manuscript. The reported results show that the presence of S in the support is important for higher propene selectivity and on-stream stability in comparison with a S-free reference material.

For industrially relevant feeds, the selectivity values are rather “standard” when considering relative low degree of propane conversion. For a feed with 33vol% C₃H₈, the selectivity is about 93% at only 18% propane conversion (Table R1). The selectivity becomes even lower, when performing PDH with a feed containing 50 % propane. Although, the degree of propane conversion is not given, I assume it is not higher than 18%. These values are not surprising when even comparing with the state-of-the-art catalysts based on metal oxides (DOI: 10.1039/d0cs01140a). In addition, the space time yield of propene formation calculated on the basis of Figure 5(a) is also not significantly higher than the corresponding values reported for metal oxide catalysts.

In terms of the rates of propene formation and deactivation, this catalyst does not show any unexpected results with industrially relevant feeds when comparing with previously tested Pt-based catalysts (Fig.8 and Table 2 in DOI: 10.1039/d0cs00814a).

Response: We really appreciate the reviewer’s patience during the two rounds of reviewing. Specially, the reviewer’s professional comments on the propane dehydrogenation are very helpful to promote the quality of our work.

In the last version of manuscript, we have supplied the conversion and selectivity results under different C₃H₈ feed (Figure S22) and we also mentioned these data in the main text (page 9). The conversion of Pt/S-C indeed decreased to only 13% at 50 vol% C₃H₈ feed. In addition, the space time yield (STY) of propene formation also decreased from 0.88 to 0.41 and 0.33 with the C₃H₈ feed increasing from 10 vol% to 33 vol% and 50 vol% (Table R1).

According to the reviewer’s comments, we tried to compare the performance of our catalyst with the reported monometallic Pt, Pt alloy, as well as metal oxide catalysts, in terms of the space time yield of propene formation, the selectivity to propene, and deactivation rate. Taking into account the difference in experimental conditions in many studies, the absolutely fair comparison between different catalysts is very challenging. To make the comparison as reasonable as possible, we restricted the test conditions as follows except for a few samples: i) the temperature of 550~600 °C; ii) the propane content of 10~50 vol%; and iii) the WHSV of 1~5 h⁻¹. We can conclude from the comparison summarized in Table R1, Figs. R1 and R2 that:

i) At high C₃H₈ feed, the Pt/S-C catalyst exhibited a lower C₃H₆ STY than some metal oxides catalysts and most Pt alloy catalysts and acquired a higher C₃H₆ STY under high WHSV (5 h⁻¹) at 10 vol% C₃H₈ feed, which is superior to most monometallic Pt and even some Pt alloy catalysts.

ii) As for the selectivity, there is little difference between Pt alloy catalysts and the Pt/S-C (above ~90% for both kinds of catalysts), while some metal oxide and monometallic Pt catalysts showed low selectivity of 75~85%.

iii) In the term of deactivation rate, the Pt/S-C is obviously in the top position with outstanding stability and even better than most Pt-alloys catalysts.

Metal oxides catalysts show considerable activity, yet suffer from the loss of oxygen under reaction conditions and thus rapid deactivation in a short time. Over the course of a catalytic cycle, due to the coke deposition, frequent regeneration operations are needed¹. We agree with reviewer that our monometallic Pt/S-C catalyst did not show unexpected C₃H₆ STY compared to the state-of-the-art catalysts metal oxide and Pt alloy catalysts. But the Pt/S-C catalyst indeed exhibited remarkable

sintering resistance and low deactivation rate, which is actually the core story of this work, as highlighted in the title, abstract, and introduction part. As we have demonstrated, the sulfur-stabilization strategy could be extended to prepare a wider range of noble metal catalysts (Pt, Ru, Rh, Os, and Ir) and applied to other high temperature reaction (such as WGS reaction, as shown in Figure S29). All these results demonstrate the validity of our concept of “Sulfur stabilizing metal nanoclusters on carbon at high temperatures”.

Inspired by the reviewer’s comments, we additionally synthesized bimetallic Pt-Sn/S-C catalyst with targets of further promoting C₃H₆ STY and selectivity meanwhile maintaining low deactivation rate, in particular at high 50 vol % C₃H₈ feed. Fortunately, we got positive results. The bimetallic Pt-Sn/S-C catalyst exhibited an enhanced activity with a conversion of 25% and also a promoted selectivity of 94% compared to monometallic Pt/S-C catalysts (13% conversion and 88% selectivity) under the identical reaction conditions. Meanwhile, the stability was well maintained with a low deactivation rate constant of 0.003 h⁻¹. The systematic optimization of the Pt-Sn/S-C catalyst and comprehensive screen of bimetallic combination of alloys may further improve the performance, but it has seriously deviated from the core concept of this work.

The above data and related discussion have been added in the revised manuscript (page 9) and supplementary materials (Fig. S28-29)

Comments: Figure R2. It is not written which feed was used. In the case of feed with 10vol% C₃H₈, 0.1 wt% Pt/S-TiO₂ deactivates significantly quicker in comparison with Pt/S-C

Response: For the Figure R2 in last response letter, 10 vol % C₃H₈ was used for the Pt/S-TiO₂ catalyst. We have updated such information.

The Pt/S-TiO₂ catalyst indeed deactivated quickly in comparison with Pt/S-C. The rapid deactivation was ascribable to the strong acid centers on the TiO₂ supports, which would lead to the formation of a significant amount of coke even under high selectivity². The coke deposition could be rapidly removed by the regeneration under oxidation atmosphere; the Pt clusters showed no sintering and the activity was recovered.

The above discussion has been added in the revised manuscript (Fig. S30).

Figure R1: Comparisons of the C_3H_6 STY, the selectivity to propene, and deactivation rate of the Pt/S-C, Pt-Sn/S-C, reported metal oxides, monometallic Pt and Pt alloy catalysts for PDH. To make the comparison as reasonable as possible, the test conditions were restricted as follows except for a few samples: the temperature of 550~600 °C, the propane content of 10~50 vol% and the WHSV of 1~5 h^{-1} . The details of these catalysts were summarized in Table S3.

Figure R2: Long-term stability test in PDH on Pt-Sn/S-C (Pt/Sn = 1). Catalytic conditions: 1 wt% Pt-Sn/S-C, atmospheric pressure, 550 °C, 50 vol% C₃H₈ feed, no Ar dilution, WHSV of propane = 3.9 h⁻¹ over 60 mg of sample.

Table R1. Comparison of the catalytic performance of the S-C supported catalysts and other reported metal oxides, monometallic Pt, and Pt alloy catalysts for PDH.

No.	Catalysts	T (°C)	WHSV (h ⁻¹)	C ₃ H ₈ Feed (vol. %)	STY(C ₃ H ₆) $kg_{C_3H_6}kg_{cat}^{-1}$	Selectivity (%)	K _d (h ⁻¹)
1	K–CrOx/Al ₂ O ₃ ³	550	4.71	40	1.17	89.5	0.467
2	K–CrOx/Al ₂ O ₃ ⁴	550	1.6	40	0.56	87	-
3	Cr ₁₀ ZrOx ³	550	9.24	40	2.14	81	1.99
4	Cr ₁₀ Zr ₉₀ /SiO ₂ ⁵	550	4.32	40	1.2	81	-
5	CrZrOx/SiO ₂ ⁶	550	3.77	40	0.73	90.8	-
6	12VOx/Al ₂ O ₃ ⁷	600	3.3	28	0.95	94	0.188
7	12V1Mg/Al ₂ O ₃ ⁸	600	3.3	28	0.86	83	0.12
8	4Zn/TiZrOx ⁹	550	4.71	40	1.28	95	0.73
9	0.05Ru/YZrOx ¹⁰	550	1.57	40	0.44	82	0.48
10	Y ₉ Zr ₉₁ Ox ⁴	550	1.6	40	0.49	90	0.422
11	m-ZrO ₂ ¹¹	550	1.89	40	0.42	86.1	-
12	2.5 wt% Cr-5 wt%Ni/Al ¹²	550	1.08	10	0.46	95	0.058
13	4.3 wt%VOx/MCM-41 ¹³	550	1.78	40	0.17	92	0.02
14	4.3 wt%VOx/(10 wt% SiO ₂ + 90 wt%Al ₂ O ₃) ¹⁴	550	1.78	40	0.21	80-94	0.05
15	Ga(i-Bu) ₃ /Al ₂ O ₃ ¹⁵	550	1.35	20	0.22	90	0.05
16	5% In ₂ O ₃ –15%Ga ₂ O ₃ –80% Al ₂ O ₃ ¹⁶	600	1.08	5	0.0046	85.5-56.8	0.067
17	Co–Al ₂ O ₃ –HAT ¹⁷	590	2.9	20	0.67	97.1	0.02
18	Co/SiO ₂ ¹⁸	550	6.38	20	0.0093	88-84	0.073
19	Zn/Al ₂ O ₃ ¹⁹	600	3	28	0.6	95	0.5
20		550	2.2	10	0.84	98	0.005
21		550	2.4	33	0.41	93	0.003
22	Pt/S-C (this work)	550	2.9	50	0.33	89	0.002
23	Mono-metallic Pt	550	5	10	2.79	96	-
24		600	2.2	10	1.09	92	0.028
25	Pt/TA10 ²	600	10	26	3.52	77	0.09

26		0.28% Pt/ND@G ²⁰	600	1.6	5	0.22	88	0.005
27		3.2 wt% Pt/CNT ²¹	600	1.6	5	0.13	75	0.011
28		Pt/Mg(Ga)(Al)O ²²	600	2.6	20	0.075	99	0.21
29		Pt/Mg(In)(Al)O ²³	600	2.6	20	0.99	98	0.13
30		0.35 wt% Pt/Al ₂ O ₃ -nanosheet ²⁴	590	9.4	16	1.08	95	0.057
31		Pt ₃ In/SiO ₂ ²⁵	600	3	<50 (C ₃ H ₈ /H ₂ =1, balance N ₂)	0.17	92	0.018
32		Pt ₃ Ga/CeAl ²⁶	600	10		4.02	99.6	0.026
33		0.42 wt% Pt0.42 wt% Sn/MgAl ₂ O ₄ ²⁷	580	2.2	10	0.95	99.5	0.004
34		PtSn/TS-1 ²⁸	600	3	16	1.47	92.5	0.033
35		PtSnIn/08Zr-Al ²⁹	600	3.3	16	1.68	98	0.097
36		0.1Pt10Cu/Al ₂ O ₃ ³⁰	520	4.0	16	0.45	90	0.001
37		4.37 wt% Pt1.55 wt% Ga/SiO ₂ ³¹	550	2	20	0.55	63.5	0.0046
38		PtSn/Al ₂ O ₃ sheet ²⁴	590	9.4	16	4.39	98	0.007
39	Pt-based alloy	PtSnAl _{0.2} /SBA-15 ³²	590	2.5	16	1.21	98.5	0.104
40		Pt-Cu/MgAl ₂ O ₄ ³³	590	6.8	47.5	1.62	87.5	0.011
41		Pt-Ag/MgAl ₂ O ₄ ³³	590	6.8	47.5	2.06	95.1	0.034
42		Pt-Sn-5/MgAl ₂ O ₄ ²⁷	580	2.4	10	0.98	98	0.003
43		0.5 wt% Pt0.9 wt% Sn/Al ₂ O ₃ (A750) ³⁴	590	5.2	16	2.35	97	0.0185
44		Pt-ZnO/Al ₂ O ₃ ¹⁹	600	3	28	0.472	94	0.045
45		PtCu/Al ₂ O ₃ ³⁰	600	4	16	1.47	90-94	0.17
46		Pt-Sn/CeO ₂ ³⁵	580	2.6	17	0.79	80-85	0.3
47		PtFe@Pt/SBA-15 ³⁶	600	3.4	26	0.97	87	0.11
48		PtZn4@S-1 ³⁷	550	3.6	25	1.63	93.2-99	0.001
49		Pt-Sn/S-C (this work)	550	3.9	50	0.88	94	0.003

References

- 1 Chen, S. *et al.* Propane dehydrogenation: catalyst development, new chemistry, and emerging technologies. *Chem. Soc. Rev.* (2021).
- 2 Jiang, F. *et al.* Propane Dehydrogenation over Pt/TiO₂-Al₂O₃ Catalysts. *ACS Catal.* **5**, 438-447 (2015).
- 3 Otroshchenko, T. P., Rodemerck, U., Linke, D. & Kondratenko, E. V. Synergy effect between Zr and Cr active sites in binary CrZrO_x or supported CrO_x/LaZrO_x: Consequences for catalyst activity, selectivity and durability in non-oxidative propane dehydrogenation. *J. Catal.* **356**, 197-205 (2017).
- 4 Otroshchenko, T. P., Kondratenko, V. A., Rodemerck, U., Linke, D. & Kondratenko, E. V. Non-oxidative dehydrogenation of propane, n-butane, and isobutane over bulk ZrO₂-based catalysts: effect of dopant on the active site and pathways of product formation. *Catal. Sci. Technol.* **7**, 4499-4510 (2017).
- 5 Han, S. *et al.* Unraveling the Origins of the Synergy Effect between ZrO₂ and CrO_x in Supported CrZrO_x for Propene Formation in Nonoxidative Propane Dehydrogenation. *ACS Catal.* **10**, 1575-1590 (2019).
- 6 Han, S. *et al.* The effect of ZrO₂ crystallinity in CrZrO_x/SiO₂ on non-oxidative propane dehydrogenation. *Appl. Catal. A: Gen.* **590**, 117350 (2020).
- 7 Liu, G., Zhao, Z.-J., Wu, T., Zeng, L. & Gong, J. Nature of the Active Sites of VO_x/Al₂O₃ Catalysts for Propane Dehydrogenation. *ACS Catal.* **6**, 5207-5214, doi:10.1021/acscatal.6b00893 (2016).
- 8 Wu, T. *et al.* Structure and catalytic consequence of Mg-modified VO_x/Al₂O₃ catalysts for propane dehydrogenation. *AIChE J.* **63**, 4911-4919 (2017).
- 9 Han, S. *et al.* Elucidating the Nature of Active Sites and Fundamentals for their Creation in Zn-Containing ZrO₂-Based Catalysts for Nonoxidative Propane Dehydrogenation. *ACS Catal.* **10**, 8933-8949 (2020).
- 10 Otroshchenko, T., Kondratenko, V. A., Rodemerck, U., Linke, D. & Kondratenko, E. V. ZrO₂-based unconventional catalysts for non-oxidative propane dehydrogenation: Factors determining catalytic activity. *J. Catal.* **348**, 282-290 (2017).
- 11 Zhang, Y. *et al.* The effect of phase composition and crystallite size on activity and selectivity of ZrO₂ in non-oxidative propane dehydrogenation. *J. Catal.* **371**, 313-324 (2019).
- 12 Li, P.-P. *et al.* The promotion effects of Ni on the properties of Cr/Al catalysts for propane dehydrogenation reaction. *Appl. Catal. A: Gen.* **522**, 172-179 (2016).
- 13 Sokolov, S., Stoyanova, M., Rodemerck, U., Linke, D. & Kondratenko, E. V. Comparative study of propane dehydrogenation over V-, Cr-, and Pt-based catalysts: Time on-stream behavior and origins of deactivation. *J. Catal.* **293**, 67-75 (2012).
- 14 Sokolov, S., Stoyanova, M., Rodemerck, U., Linke, D. & Kondratenko, E. Effect of support on selectivity and on-stream stability of surface VO_x species in non-oxidative propane dehydrogenation. *Catal. Sci. Technol.* **4**, 1323-1332 (2014).
- 15 Szeto, K. C. *et al.* A strong support effect in selective propane dehydrogenation catalyzed by Ga (i-Bu)₃ grafted onto γ -alumina and silica. *ACS Catal.* **8**, 7566-7577 (2018).
- 16 Tan, S. *et al.* Propane dehydrogenation over In₂O₃-Ga₂O₃-Al₂O₃ mixed oxides. *ChemCatChem* **8**, 214-221 (2016).
- 17 Dai, Y. *et al.* γ -Al₂O₃ sheet-stabilized isolate Co²⁺ for catalytic propane dehydrogenation. *J. Catal.* **381**, 482-492 (2020).
- 18 Estes, D. P. *et al.* C-H Activation on Co, O Sites: Isolated Surface Sites versus Molecular Analogs. *J. Am. Chem. Soc.* **138**, 14987-14997 (2016).
- 19 Liu, G. *et al.* Platinum-Modified ZnO/Al₂O₃ for Propane Dehydrogenation: Minimized Platinum Usage and Improved Catalytic Stability. *ACS Catal.* **6**, 2158-2162, doi:10.1021/acscatal.5b02878 (2016).
- 20 Liu, J. *et al.* Origin of the robust catalytic performance of nanodiamond-graphene-supported Pt nanoparticles used in the propane dehydrogenation reaction. *ACS Catal.* **7**, 3349-3355 (2017).
- 21 Liu, J. *et al.* Defect-driven unique stability of Pt/carbon nanotubes for propane dehydrogenation. *Appl. Surf. Sci.* **464**, 146-152 (2019).
- 22 Siddiqi, G., Sun, P., Galvita, V. & Bell, A. T. Catalyst performance of novel Pt/Mg (Ga)(Al) O catalysts for alkane dehydrogenation. *J. Catal.* **274**, 200-206 (2010).
- 23 Xia, K., Lang, W.-Z., Li, P.-P., Yan, X. & Guo, Y.-J. The properties and catalytic performance of PtIn/Mg (Al) O catalysts for the propane dehydrogenation reaction: Effects of pH value in preparing Mg (Al) O supports by the co-precipitation method. *J. Catal.* **338**, 104-114 (2016).
- 24 Shi, L. *et al.* Al₂O₃ Nanosheets Rich in Pentacoordinate Al³⁺ Ions Stabilize Pt-Sn Clusters for Propane Dehydrogenation. *Angew. Chem. Int. Ed.* **54**, 13994-13998 (2015).
- 25 Zha, S. *et al.* Identification of Pt-based catalysts for propane dehydrogenation via a probability analysis. *Chem. Sci.* **9**, 3925-3931 (2018).
- 26 Wang, T. *et al.* Effects of Ga doping on Pt/CeO₂-Al₂O₃ catalysts for propane dehydrogenation. *AIChE J.* **62**, 4365-4376 (2016).
- 27 Zhu, H. *et al.* Sn surface-enriched Pt-Sn bimetallic nanoparticles as a selective and stable catalyst for propane dehydrogenation. *J. Catal.* **320**, 52-62 (2014).

- 28 Li, J. *et al.* Size effect of TS-1 supports on the catalytic performance of PtSn/TS-1 catalysts for propane
dehydrogenation. *J. Catal.* **352**, 361-370 (2017).
- 29 Long, L.-L. *et al.* The comparison and optimization of zirconia, alumina, and zirconia-alumina supported PtSnIn
trimetallic catalysts for propane dehydrogenation reaction. *J. Ind. Eng. Chem.* **51**, 271-280 (2017).
- 30 Sun, G. *et al.* Breaking the scaling relationship via thermally stable Pt/Cu single atom alloys for catalytic
dehydrogenation. *Nat. Commun.* **9**, 4454, doi:10.1038/s41467-018-06967-8 (2018).
- 31 Searles, K. *et al.* Highly productive propane dehydrogenation catalyst using silica-supported Ga–Pt nanoparticles
generated from single-sites. *J. Am. Chem. Soc.* **140**, 11674-11679 (2018).
- 32 Fan, X. *et al.* Dehydrogenation of propane over PtSnAl/SBA-15 catalysts: Al addition effect and coke formation
analysis. *Catal. Sci. Technol.* **5**, 339-350 (2015).
- 33 Ren, G.-Q. *et al.* Effect of group IB metals on the dehydrogenation of propane to propylene over anti-sintering
Pt/MgAl₂O₄. *J. Catal.* **366**, 115-126 (2018).
- 34 Jang, E. J., Lee, J., Jeong, H. Y. & Kwak, J. H. Controlling the acid-base properties of alumina for stable PtSn-
based propane dehydrogenation catalysts. *Appl. Catal. A: Gen.* **572**, 1-8 (2019).
- 35 Xiong, H. *et al.* Thermally stable and regenerable platinum–tin clusters for propane dehydrogenation prepared by
atom trapping on ceria. *Angew. Chem. In. Ed.* **129**, 9114-9119 (2017).
- 36 Cai, W. *et al.* Subsurface catalysis-mediated selectivity of dehydrogenation reaction. *Sci. Adv.* **4**, eaar5418 (2018).
- 37 Sun, Q. *et al.* Subnanometer Bimetallic Platinum-Zinc Clusters in Zeolites for Propane Dehydrogenation. *Angew.
Chem. In. Ed.* **59**, 19450-19459, doi:10.1002/anie.202003349 (2020).

REVIEWERS' COMMENTS

Reviewer #1 (Remarks to the Author):

The authors have clarified my concerns. However, it may be useful if the authors added Table R1 to the manuscript to support high performance of their catalysts.